# Identification of Large Japanese field mouse *Apodemus speciosus* food plant resources in an industrial green space using DNA metabarcoding

Taichi Fujii[ID][1][☯]*, Hirokazu Kawamoto[2][☯], Tomoyasu Shirako[3][☯], Masatoshi Nakamura[3][☯], Motoyasu Minami[1][☯]

**1** Bioscience and Biotechnology, Chubu University, Kasugai, Aichi, Japan, **2** Ohki Pharmaceutical Co., Ltd., Tokyo, Japan, **3** Institute of Environmental Ecology, IDEA Consultants, Inc., Yaizu, Shizuoka, Japan

☯ These authors contributed equally to this work.
* fujii_t@fsc.chubu.ac.jp

## Abstract

DNA metabarcoding was employed to identify the food plant resources of the Large Japanese field mouse *Apodemus speciosus* inhabiting an artificial green space on reclaimed land on the Chita Peninsula in Aichi Prefecture, Central Japan, from 2012 to 2014. DNA metabarcoding was performed using high-throughput sequencing of partial *rbcL* sequences extracted from fecal samples collected in the study area. The obtained sequences, which were analyzed using a constructed local database, revealed that a total of 72 plant taxa were utilized as food plant resources by *A. speciosus*. Of these plant taxa, 43 could be assigned to species (59.7%), 16 to genus (22.2%), and 13 to family (18.1%). Of the 72 plant taxa identified in this study, the dominant families throughout all collection periods were Lauraceae (81.0% of 100 fecal samples), followed by Fagaceae (70.0%), Rosaceae (68.0%), and Oleaceae (48.0%). Fifty of the 72 plant taxa identified as food plant resources were woody plants. The findings showed that DNA metabarcoding using a local database constructed from the National Center for Biotechnology Information (NCBI) database and field surveys was effective for identifying the dominant food plants in the diet of *A. speciosus*. The results of this study provided basic information that can be applied to the formulation and implementation of management and conservation strategies for local wildlife.

## Introduction

Green spaces in urban areas serve as critical habitats for local wildlife and contribute to the conservation of biodiversity. These areas help to maintain natural environments that might otherwise be lost due to urbanization, providing suitable habitat for a range of species, even within a city [1–4]. In addition, green spaces in and near urban areas play important roles in harboring biodiversity and promoting human well-being [5,6]. As a result, the importance of proper conservation and management of urban green spaces is increasingly being realized. In

**Data availability statement:** All sequencing files are available from the NCBI database (BioProject number PRJDB17374). And, All result of this study are within the manuscript and its Supporting Information files.

**Funding:** Funding: This work was supported by subsidies for environmental activities and education from Aichi Forest and Greenery in Aichi Prefecture, Japan (Minami M, Grant Number 24-122, 25-57, and 26-105, https://www.pref.aichi.jp/soshiki/kankyokatsudo/mori-kankyou.html), and the Challenge Site at Chubu University (Minami M, Grant Number 12-9168, 13-9168, and 14-9168, https://www.chubu.ac.jp/english/) The funders had no role in study design, data collection and analysis, decision to publish, or preparation of the manuscript.

**Competing interests:** The authors have declared that no competing interests exist.

addition, the United Nations Convention on Biological Diversity (CBD) has proposed that at least 30% of the Earth's land and sea areas should be conserved by 2030 (also referred to as the "30-by-30" initiative) to protect the natural environment [7]. To achieve this goal, several countries have designated areas for effective biodiversity conservation as Other Effective area-based Conservation Measures (OECMs) [8–12]. In Japan, to achieve the 30-by-30 target, areas such as urban green spaces and artificial green spaces, where biodiversity is maintained through the activities of local people, have been designated and supported as OECMs [10]. Previous studies have shown that species such as the red fox (*Vulpes vulpes*) [13–15], raccoon dog (*Nyctereutes procyonoides*) [14,16], northern goshawk (*Accipiter gentilis*) [17], and the Japanese scops owl (*Otus semitorques*) [18] have been confirmed to inhabit and breed in artificial green spaces in urban areas. Considering that wildlife populations in Japan have been threatened by extensive habitat destruction, mainly due to urbanization, artificial green spaces with habitats for small mammals in urban areas have contributed positively towards maintaining viable populations of wildlife [14,19]. Residents of urban areas are aware of the value of artificial green spaces in urban areas for wildlife and wish to protect and coexist with wild animals [20]. Consequently, artificial green spaces are being assessed for their utility as wildlife habitats in Japan [4,10,14,21].

Of the small mammals distributed in Japan, the Large Japanese field mouse *Apodemus speciosus* is an endemic species that is widely distributed throughout the Japanese Archipelago, except the Ryukyu Islands at the southwestern-most tip of Japan [22]. This species is omnivorous and forages on forest resources such as seeds, insects, and other small invertebrates [22]. Additionally, it also plays a role as a food resource for carnivorous animals [23]. Thus, *A. speciosus* forms an important connection between vegetation and predators, and has a significant influence on forest regeneration through seed dispersal [24,25]. Consequently, it is considered that this species plays a crucial role in maintaining forest ecosystems. While this species typically inhabits forest environments, it has also colonized artificially created green spaces in urban areas [14,19]. In order to maintain ecosystem health in the artificial green spaces of urban areas, it is necessary to maintain populations of small mammals such as *A. speciosus*, which are preyed upon by carnivorous animals [26]. Since the population of *A. speciosus* is significantly affected by the abundance of food resources, such as acorns [27], proper vegetation management could potentially contribute to maintaining healthy populations of this species. The survey of plant food resources utilized by *A. speciosus* could therefore provide important information for maintaining and managing urban green spaces, especially within the context of the food pyramid dynamics in these environments.

Several methods have been employed to identify the food resources utilized by wildlife, including identification of gastric or feces contents by dissection [28–31] and direct observations of foraging behavior [32,33]. Although the diet of *A. speciosus* has been studied extensively, as a small rodent with a head-body length of 80–140 mm [22], identifying the food resources utilized by this species through examining fragmented plant remains in stomach contents is challenging [34–36]. Recently, DNA metabarcoding using high-throughput sequencing (HTS) has been used to identify the food resources of *A. speciosus* with higher sensitivity [37–39]. However, the results obtained using the chloroplast *trnL* P6 loop intron region yielded results with a low resolution; specifically, only 2.9% [37] and 15.7% [38] of detected plant taxa could be identified to species. Consequently, high-resolution identification of *A. speciosus* food plant resources at the species level remains unclear. In addition, while several studies have examined the food plant resources of *A. speciosus* in natural habitats using DNA metabarcoding [37–39], the food plant resources in artificial green spaces in urban areas has only been reported using cloning methods [14] and not by DNA metabarcoding. In general, the chloroplast *trnL* P6 loop intron, *rbcL* and *matK* regions, and the nuclear ITS2

region have been used for plant identification using DNA metabarcoding, and integrating the obtained results from these regions has been shown to improve resolution and detection rates [40]. It has been reported that integrating *rbcL*, which has broad applicability for plant identification, with ITS2, which provides high-resolution species identification but lacks broad applicability, is suitable for species-level plant identification [40]. However, analyzing multiple regions increases the cost of the analysis. In addition, DNA metabarcoding has been shown to improve resolution by referencing local databases [41,42]. For this reason, we constructed a local database from vegetation survey results obtained by an environmental consulting company, and fitted these data to study sites. Based on vegetation surveys conducted by an environmental consulting company in 2001, and by us in this study, a total of 796 species were identified in the study area (S1 Table). By searching txidXXXX[Organism:-exp] AND YYYY[All Fields] (XXXX: Taxonomy ID, YYYY: DNA region name) on the NCBI website, we counted the number of hits. These keywords used for the DNA region names in the search were *rbcL*, transcribed+spacer+2 (ITS2), and tRNA-Leu (*trnL* p6 loop). As a result of searching the number of registrations for these 796 species in the *rbcL*, ITS2, and *trnL* p6 loop regions on NCBI, it was found that *rbcL* had 722 species, ITS2 had 680 species, and *trnL* p6 loop had 631 species (searched on November 25, 2024). In this study, we used *rbcL*, which has higher classification resolution for plant species identification than the *trnL* P6 loop and has primers that are more widely applicable than ITS2 [65,67,43], and has a higher number of registrations to NCBI, to identify the dominant food plant resources of *A. speciosus* by DNA metabarcoding utilizing a local database. We consider that the results of this study provide basic information that can be applied to the development of effective management and conservation strategies for local wildlife. In addition, the findings also demonstrate the utility of constructing a local database for high-resolution analysis of food resources using DNA metabarcoding.

## Materials and methods

### Trapping rodents and fecal sample collection

The study site is located in industrial green spaces on reclaimed land on the Chita Peninsula in Aichi Prefecture, Central Japan (Fig 1). Since the industrial area opened in the 1970s, it has been separated from the surrounding area by Ise Bay to the west and Chita Industrial Road, which is more than 30 m wide, to the east. Based on vegetation surveys conducted by an environmental consulting company in 2001, and by us in this study, the vegetation at the site consists of a mixture of evergreen broad-leaved trees and deciduous broad-leaved trees, which were either planted when the industrial area was built or which naturally colonized the area from the surrounding environment. The dominant species in the study site were *Cinnamomum camphora*, *Lithocarpus edulis*, *Pterocarya stenoptera*, *Ligustrum lucidum*, and *Elaeocarpus zollingeri* var. *zollingeri* (see S1 Table for details). The climate of the study area is characterized by having a mean annual temperature and precipitation of 16.1°C and 1450.3 mm, respectively [44]. This study was conducted after obtaining a license for capturing *A. speciosus* from the Aichi Prefectural Government (Ochi No. 3–1) and was performed in full compliance with the Guidelines for Animal Treatment proposed by the Mammal Society of Japan, with all efforts made to minimize animal suffering. Trapping was conducted at approximately 2-month intervals from May 2012 to November 2014. Twelve line transects were established along the footpaths within the industrial green spaces, with a total of 308 Sherman-type live traps (6.5 × 5.5 × 16.0 cm, H. B. Sherman Traps, FL, USA) were set such that there were two traps at each point along each transect at 10-meter intervals. The bait used for capturing *A. speciosus* consisted of a mixture of barley (*Hordeum vulgare* subsp.

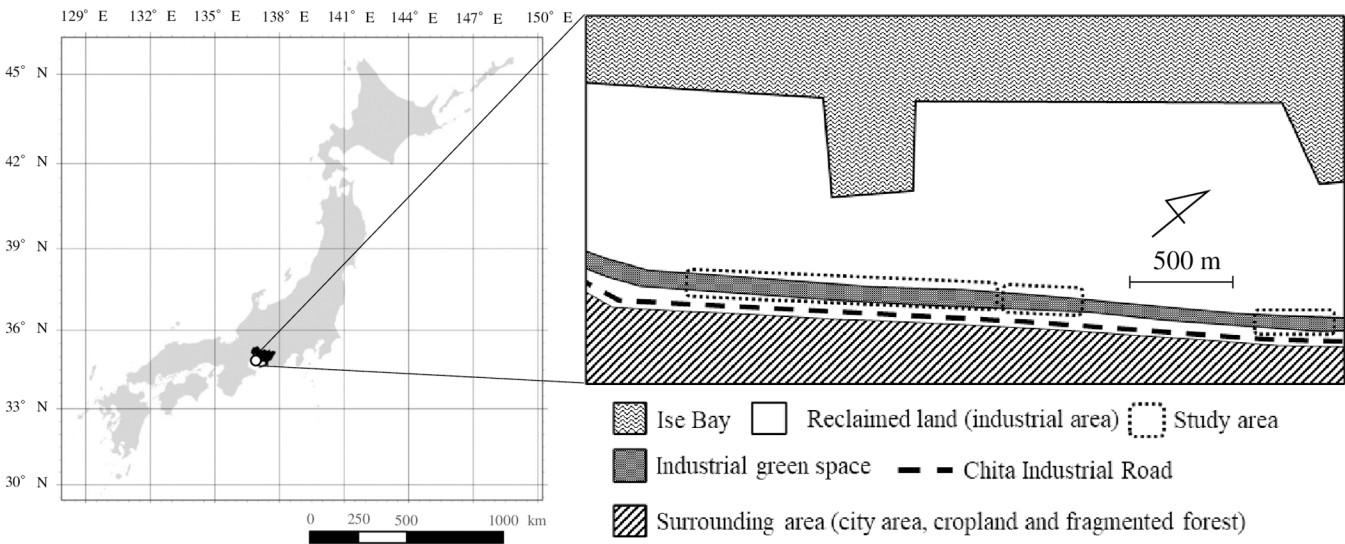

**Fig 1. Location of the industrial green spaces investigated in this study and an overview of the surroundings.** The industrial green spaces were located between the area of reclaimed land and the Chita Industrial Road. Dotted lines indicate areas where the Large Japanese field mice (*Apodemus speciosus*) were trapped. Republished from digital national land information (https://nlftp.mlit.go.jp/ksj/) under a CC BY license, with permission from Geospatial Information Division, Ministry of Land, Infrastructure, Transport and Tourism, the government of Japan, original copyright 3 Feb. 2025.

*vulgare*) and walnuts (*Juglans regia*) (15:1 w/w). We live-captured at total of 100 *A. speciosus* individuals at the study site (spring: 22 from March to May, summer: 35 from June to August, autumn: 32 from September to November, winter: 11 from December to February). Trapped individuals were released immediately after capture at the site of capture and all of the feces in the Sherman trap were collected for total DNA extraction. To minimize artificial contamination, each fecal sample was quickly placed into a 1.5 mL tube using tweezers that had been thoroughly disinfected and washed with ethanol. The fecal samples thus collected were transported to the laboratory at room temperature on that day and then stored at −20°C until total genomic DNA extraction.

### Partial sequencing of *rbcL* gene in fecal samples

Differences in the weights of the fecal samples analyzed may affect the number of plant species detected in a single fecal sample. To ensure uniform weight of a single fecal sample, dry weight (ca. 60 mg) was used for analysis. All fecal samples were dried overnight at 60°C before performing DNA extraction as described in Shirako et al. [45]. Total DNA was isolated from the dried feces using a DNeasy Plant Mini Kit (Qiagen, Hilden, Germany) and purified using a Geneclean Spin Kit (MP-Biomedicals, Santa Ana, CA, USA). In this study, the identification of food plant species in feces was performed by DNA metabarcoding using HTS of the partial sequences of the *rbcL* region in chloroplast DNA (*rbcL* F3R3) [46]. Since amplicon sequencing using Illumina HTS platforms typically decreases nucleotide diversity, adjacent DNA sequences on the flow cell can be misrecognized and low sequencing quality scores can result [47]. Therefore, in order to compensate for the potential reduction in nucleotide diversity, we used frame-shifting primers for the initial PCR [47]. The initial PCR targeting the *rbcL* F3R3 region was conducted using KAPA HiFi HotStart ReadyMix (Kapa Biosystems, Wilmington, Massachusetts, MA, USA), with each 12.0 μL total reaction mixture containing 2.0 μL of template DNA, 6.0 μL of KAPA HiFi HotStart ReadyMix, 0.29 μmol/L of each primer pair (see S2 Table), and ultrapure water to make up the

final volume. After initial denaturation at 95°C for 3 min, samples were subjected to 35 cycles of denaturation at 98°C for 20 s, annealing at 56°C for 15 s, and extension at 72°C for 30 s, with a final extension step at 72°C for 5 min. The initial PCR products of *rbcL* F3R3 were purified using an Agencourt AMPure XP kit (Beckman Coulter, Fullerton, CA, USA). To construct the DNA libraries for the second PCR, i7 and i5 indexes (Illumina Inc., San Diego, CA, USA) were used to identify each sample, and P5 and P7 adapters (Illumina Inc.) for Miseq sequencing were ligated to the purified initial PCR products. The second PCR targeting the *rbcL* F3R3 region was conducted using KAPA HiFi HotStart ReadyMix (Kapa Biosystems). Each 24.0 μL reaction mixture contained 2.0 μL of 1/10 diluted initial PCR products, 12.0 μL of KAPA HiFi HotStart ReadyMix (Kapa Biosystems), 0.6 μmol/L of forward and reverse index primers (see S3 Table), and ultrapure water to make up the final volume. After initial denaturation at 95°C for 3 min, the samples were subjected to 12 cycles of denaturation at 98°C for 20 s and extension at 72°C for 15 s, and a final extension step at 72°C for 5 min. The DNA libraries of *rbcL* F3R3 were then purified using an Agencourt AMPure XP kit (Beckman Coulter). Library quantification was performed by qPCR using a GenNext NGS library quantification kit (TOYOBO, Osaka, Japan). Amplicons were combined in approximately equimolar concentrations to produce a single DNA library of all extracts for sequencing. Sequencing was then performed using a MiSeq Reagent Kit v3 (600-cycle format; Illumina Inc.) on an Illumina MiSeq sequencer (Illumina Inc.) by IDEA Consultants, Inc. (Yaizu, Japan), following the manufacturer's instructions.

## Sequence data analysis

Using the bcl2fastq v2.18 program (Illumina Inc.), the raw MiSeq data were first converted to FASTQ files, which were then demultiplexed using the clsplitseq function implemented in Claident [48]. In this study, the demultiplexed FASTQ files were analyzed using the amplicon sequence variant (ASV) method implemented in the DADA2 v1.10.1 package [49] using the R statistical software package [50]. As part of the quality filtering process, both forward and reverse sequences of *rbcL* F3R3 were trimmed to 200 bases. Trimming was performed based on visual inspection of the quality score distribution using the filterAndTrim function of DADA2 [49]. After trimming, forward and reverse sequences were combined using the mergePairs function of DADA2 [49], and chimeric DNA sequences were removed using the removeBimeraDenovo function of DADA2 [49]. The fecal samples collected in the field may be contaminated with plants other than those foraged by the rodents. To address this issue, it has been reported that removing approximately 1% of low-frequency sequences from each fecal sample yields reliable feeding data [51]. While this step may also remove sequences derived from the foraging materials [51], the objective of this study was not to examine the differences in food resources among individual *A. speciosus* living in urban green spaces, but rather to analyze the dominant plant species utilized by the *A. speciosus* population. Therefore, in this study, low-frequency ASVs, defined as those representing fewer than 1.0% of the total number of sequences in each fecal sample, were excluded from the DNA metabarcoding analysis. To normalize the number of DNA sequences in *rbcL* F3R3, rarefaction curves were calculated using the rarecurve function implemented in the vegan package ver. 2.5–5 [52] in R [50] (S1 Fig). Using the calculated rarefaction curves, 1,000 *rbcL* F3R3 DNA sequences from each fecal sample were normalized using the rrarefy function in the vegan R package [52] to determine the ASVs for plant species identification.

## Construction of partial *rbcL* database for the fecal samples collected in the study area

Based on vegetation surveys conducted by an environmental consulting company in 2001, and by us in this study, a total of 796 species were identified in the study area (S1 Table). The

*rbcL* F3R3 database of these 796 species, as well as barley (*H. vulgare* subsp. *vulgare*) and walnut (*J. regia*), which were used as bait for capturing mice, was constructed as follows. After obtaining the NCBI taxonomy IDs of the 798 species from the NCBI Taxonomy Database, we retrieved the NCBI GI numbers containing the *rbcL* sequences of 798 species required to construct a local database. This was done by using the taxonomy IDs of the 798 species and *rbcL* as keywords with the clretrievegi function implemented in Claident [48] (S1 Table). The NCBI GI numbers were then converted to GenBank format using the pgretrieveseq function implemented in Claident [48]. The FASTA format data were converted from these GenBank format data using the extractfeat function in the EMBOSS package [53]. Next, the obtained sequences were trimmed to match the *rbcL* F3R3 region (262 bp) using MEGA6 [54]. As a result, 147 of the 798 plant species were not registered in the NCBI database (S1 Table). It is important to note here that species identification for newly registered DNA sequences in the database is self-reported by the submitter which can occasionally lead to sequences being associated with misidentified species [55]. To achieve high-resolution identification of food plant resources for *A. speciosus* at the study site, we sequenced the *rbcL* F3R3 region of 104 dominant plant species at the study site using the following methods. Total DNA was extracted from plant specimens (ca. 0.8 cm$^2$) using a DNeasy Plant Mini Kit (Qiagen) and purified using a Geneclean Spin Kit (MP-Biomedicals). PCR was then performed using a reaction mixture of 50 μL containing 1 unit of MightyAmp DNA Polymerase Ver. 2 (Takara, Shiga, Japan) and 0.32 μM of each primer according to the manufacturer's instructions. The *rbcL* F3R3 amplicon was sequenced using the forward and reverse primers used in the aforementioned initial PCR for DNA metabarcoding using HTS to identify the plant species in the fecal samples [46]. PCR amplification was performed using a DNA Thermal Cycler (GeneAmp PCR System 9700, Applied Biosystems, Foster City, CA) using an initial denaturation step of 98°C for 2 min, followed by 30 cycles of denaturation at 98°C for 10 s, annealing at 60°C for 15 s, and extension at 68°C for 20 s. All PCR products were purified using a QIAquick PCR Purification Kit (Qiagen) and subjected to dye-terminator cycle sequencing using DTCS Quick Start Mix (Beckman Coulter) and an automatic sequencer (CEQ 2000XL, Beckman Coulter). Next, sequence alignment was performed using the ClustalW algorithm implemented in MEGA6 [54]. Finally, the *rbcL* F3R3 sequences obtained from the NCBI database and the 104 plant species collected at the study site were combined to construct a local *rbcL* F3R3 database consisting of 651 plant species using the makeblastdb function implemented in NCBI's BLAST+ software package [56] (S1 Table).

## Homology search and identification of food plant resources

DNA metabarcoding using HTS to identify the plant species in fecal samples was conducted using a combination of our local *rbcL* F3R3 database and the NCBI database. Fig 2 shows a flowchart of the steps used to identify *A. speciosus* food plant resources in this study. To identify food plant species, homology searches were performed by comparing the ASVs obtained from each fecal sample against the local *rbcL* F3R3 database using the BLASTn function implemented in NCBI's BLAST+ version 2.6.0+ package [56]. If the percent identity from the local *rbcL* F3R3 database was less than 98%, these ASVs were subjected to a homology search using all of the published sequences deposited in the NCBI database. To ensure that the identification of food plant resources was sufficiently robust, DNA sequences of *rbcL* F3R3 with a percent identity of less than 98% in either database were excluded from further analysis. In addition, the ASVs that were identified as belonging to either barley (*H. vulgare* subsp. *vulgare*) or walnut (*J. regia*), which were used as bait, were also excluded from further analysis. In the event that a given sequence was identified as belonging to two or more taxa with the same score, that sequence was assigned to the highest taxonomic level that included both of those

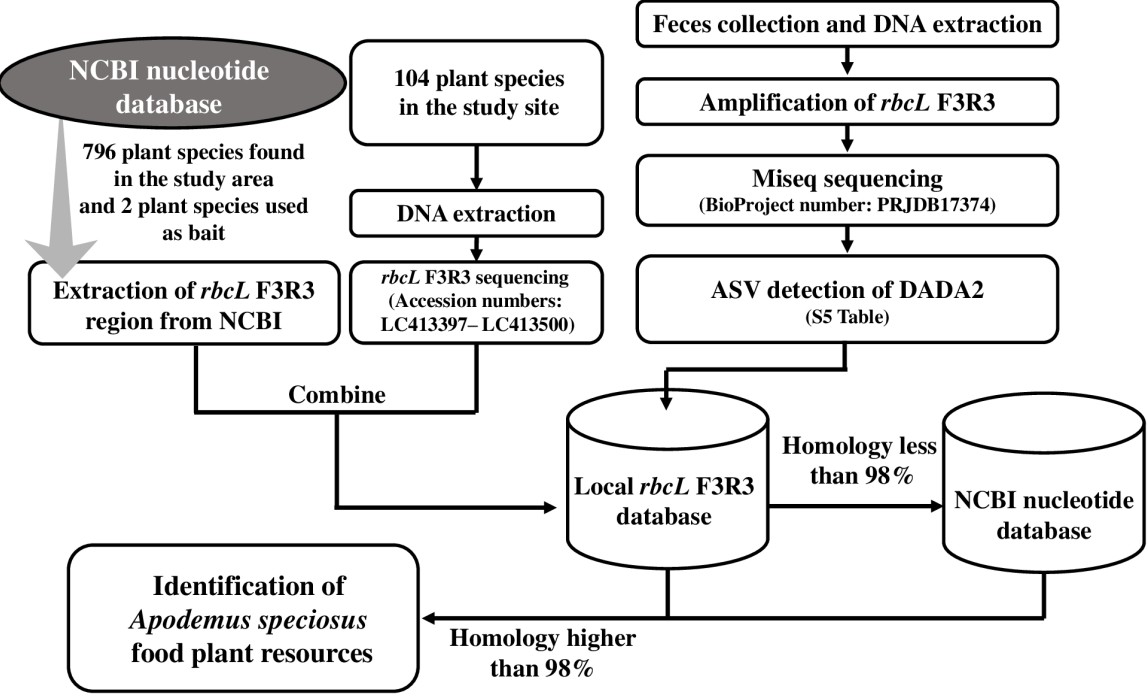

**Fig 2. Flowchart showing the process used to identify the food plant resources of the Large Japanese field mice (*Apodemus speciosus*) in this study.**

taxa. To assess the diet frequency of occurrence of each taxon (i.e., species, genus or family), the total number of plant taxa identified at each level was divided by the total number of fecal samples. In addition, we summarized taxonomic resolution in this study relative to dividing the total number of plant taxa identification at species, genus, and family level by the total number of plant taxa.

## Statistical analysis

To compare four seasonal differences (spring: March to May, summer: June to August, autumn: September to November, winter: December to February) in the diet, pairwise Bray-Curtis and Jaccard indices were calculated for differences in plant resources in sampled feces using relative read abundance data and binary (0/1) occurrence rate data, respectively. The obtained matrices were used to construct a non-metric multidimensional scaling (NMDS) plot, using the vegan package [52] in R [50]. Due to the high stress value (> 0.2) in the 2D NMDS plot, a 3D NMDS plot was constructed using the scatterplot3d package [57]. Differences in diet between four seasons were assessed using PerMANOVAs with 10,000 permutations (alpha level = 0.05) [52]. In addition, to compare seasonal food plant diversity, coverage-based rarefaction and extrapolation curves were generated using the Shannon diversity index. The coverage-based R/E curve compares species richness across assemblages based on samples of equal completeness (equal coverage) [58–60]. This method allows for comparison of diversity within a community by normalizing bias due to sample coverage [58–60]. The coverage-based R/E curve was plotted using the iNEXT package ver. 2.0.20 [61,62] in R [50]. The number of plant taxa identified per fecal sample in spring, summer, autumn, and winter, was compared by the Steel-Dwass test (alpha level = 0.05) with the pSDCFlig function in the NSM3 package ver. 1.16 [63] in R [50].

## Results

### Construction of local *rbcL* F3R3 database

The 5934 DNA sequences in the local *rbcL* F3R3 database were obtained from 651 plant species in the NCBI database and 104 predominant plant species collected at the study site (S1 Table). The *rbcL* F3R3 sequences obtained from the 104 predominant plant species at the study site were registered with the DNA Data Bank of Japan under accession numbers LC413397–LC413500. Of the 782 unique sequences in the local *rbcL* F3R3 database, 631 (80.7%) were species-specific; 96 (12.3%) were the same at the genus level, but not at the species level; 44 (5.6%) were the same at the family level, but not at the genus level; and 11 (1.4%) could not be assigned, even at the family level.

In this local *rbcL* F3R3 database, walnut (*J. regia*), which was used as bait, and *Pterocarya rhoifolia* and *P. stenoptera*, which were found at the study site, were excluded from further analysis because it is impossible to determine whether these DNA sequences could be attributed to bait or plants growing at the study site because they grew at the study area.

### Sequence data processing

The *rbcL* F3R3 region was successfully amplified from all 100 fecal samples. Sequencing of the *rbcL* F3R3 region yielded a total of 8,895,012 DNA sequences (after quality filtering and removing chimeric sequences) (S4 Table). In this study, the *rbcL* F3R3 sequence data were normalized to 1,000 DNA sequences per fecal sample, and ASVs were identified from a total of 100,000 DNA sequences, resulting in the distinction of 201 ASVs (S5 Table). The rarefaction curves for all samples plateaued sufficiently at 1,000 reads, confirming that no ASVs were lost due to this processing. All nucleotide sequence data were deposited in the DDBJ database (BioProject number PRJDB17374).

### Identification of food plant resources

Based on the results of homology searches using the local *rbcL* F3R3 database, 185 out of 201 ASVs (92.0%) were identified with more than 98% identity. For the 16 ASVs (8.0%) with less than 98% identity in the local *rbcL* F3R3 database, a homology search was conducted using the NCBI database. The results indicated that 12 of these 16 ASVs were identified as belonging to seven taxa with more than 98% identity using the NCBI database (S5 Table). Among these seven plant taxa, *Broussonetia* spp., *Camptotheca acuminate*, and *Croton tiglium*, which are commonly planted for green space maintenance in Japan, were considered as food plant resources for *A. speciosus.* Conversely, *Gentiana nipponica, Sieversia pentapetala*, and *Vaccinium* spp., which grow in the alpine meadow zone of Japan, as well as barley (*H. vulgare* subsp. *vulgare*), which was used as bait in this study, were not considered food plant resources for *A. speciosus.* The remaining four ASVs had less than 98% identity with sequences in both our local *rbcL* F3R3 database and the NCBI database, and were therefore excluded from this study.

Total results of the DNA metabarcoding analysis of food plant sequences are summarized in Table 1. Using the local *rbcL* F3R3 database, 192 ASVs were identified with more than 98% identity, corresponding to 72 plant taxa in 43 families. Of these, 43 (59.7%) were identified to the species level, 16 (22.2%) to the genus level, and 13 (18.1%) to the family level (Table 1). Of the 43 plant families identified using the local *rbcL* F3R3 database, the dominant families during all collection periods were Lauraceae (81.0% of 100 fecal samples), followed by Fagaceae (70.0%), Rosaceae (68.0%), and Oleaceae (48.0%). The dominant food plant taxa identified were *Cinnamomum* spp. (79.0%), followed by Fagaceae-1 (55.0%), Rosaceae (53.0%), Oleaceae-2 (47.0%), and *Quercus* spp. (39.0%) (Table 1); numbers following family names serve as identifiers to distinguish different ASVs within the same taxonomic group (see Table 1 for details). A total of 50 of the 72 plant taxa identified as food plant resources were woody plants (Table 1). To visualize seasonal changes in diet, we plotted the results of NMDS

**Table 1. Food plant taxa identified using DNA metabarcoding of the partial *rbcL* region.**

| Family name | Identified plant taxa | Identified plant names | identi-fied tax-onomic level | Woody plant or Herbaceous plant | Data-base [1] | No. of fecal samples | | | | all (n=100) |
|---|---|---|---|---|---|---|---|---|---|---|
| | | | | | | Season | | | | |
| | | | | | | Spring (Mar. to May) (n=22) | Summer (Jun. to Aug.) (n=35) | Autumn (Sept. to Nov.) (n=32) | Winter (Dec. to Feb.) (n=11) | |
| Adoxaceae | Adoxaceae | *Sambucus racemosa* subsp. *sieboldiana, Viburnum awabuki* | Family | Woody plant | L | 1 (4.5%) | 1 (2.9%) | 6 (18.8%) | 2 (18.2%) | 10 (10%) |
| Anacardia-ceae | *Toxicodendron* spp. | *Toxicodendron sylvestre, Toxicodendron succedaneum* | Genus | Woody plant | L | 1 (4.5%) | 4 (11.4%) | 0 (0%) | 0 (0%) | 5 (5%) |
| Apiaceae | Apiaceae | *Osmorhiza aristata* var. *aristata, Torilis scabra, Torilis japonica* | Family | Herbaceous plant | L | 0 (0%) | 0 (0%) | 1 (3.1%) | 0 (0%) | 1 (1%) |
| Apocyna-ceae | *Nerium oleander* var. *indicum* | *Nerium oleander* var. *indicum* | Species | Woody plant | L | 0 (0%) | 1 (2.9%) | 1 (3.1%) | 0 (0%) | 2 (2%) |
| Apocyna-ceae | *Trachelosper-mum asiaticum* var. *asiaticum* | *Trachelospermum asiaticum* var. *asiaticum* | Species | Woody plant | L | 0 (0%) | 1 (2.9%) | 0 (0%) | 0 (0%) | 1 (1%) |
| Aquifolia-ceae | *Ilex* spp. | *Ilex crenata, Ilex pedunculosa* | Genus | Woody plant | L | 8 (36.4%) | 6 (17.1%) | 2 (6.3%) | 0 (0%) | 16 (16%) |
| Aquifolia-ceae | *Ilex rotunda* | *Ilex rotunda* | Species | Woody plant | L | 1 (4.5%) | 0 (0%) | 1 (3.1%) | 0 (0%) | 2 (2%) |
| Araliaceae | Araliaceae | *Dendropanax trifidus, Hedera rhombea* | Family | Woody plant | L | 2 (9.1%) | 3 (8.6%) | 2 (6.3%) | 1 (9.1%) | 8 (8%) |
| Arecaceae | *Trachycarpus fortunei* | *Trachycarpus fortunei* | Species | Woody plant | L | 0 (0%) | 1 (2.9%) | 0 (0%) | 0 (0%) | 1 (1%) |
| Asteraceae | Asteraceae | *Centipeda minima, Petasites japonicus* var. *japonicus, Symphyotrichum subulatum* var. *subulatum, Taraxacum officinale* | Family | Herbaceous plant | L | 7 (31.8%) | 2 (5.7%) | 5 (15.6%) | 1 (9.1%) | 15 (15%) |
| Asteraceae | *Farfugium japonicum* var. *japonicum* | *Farfugium japonicum* var. *japonicum* | Species | Herbaceous plant | L | 5 (22.7%) | 0 (0%) | 3 (9.4%) | 0 (0%) | 8 (8%) |
| Asteraceae | *Artemisia indica* var. *maximowiczii* | *Artemisia indica* var. *maximowiczii* | Species | Herbaceous plant | L | 0 (0%) | 1 (2.9%) | 0 (0%) | 0 (0%) | 1 (1%) |
| Cannaba-ceae | *Aphananthe aspera* | *Aphananthe aspera* | Species | Woody plant | L | 2 (9.1%) | 5 (14.3%) | 7 (21.9%) | 0 (0%) | 14 (14%) |
| Cannaba-ceae | Cannabaceae | *Aphananthe aspera, Celtis sinensis* | Family | Woody plant | L | 2 (9.1%) | 2 (5.7%) | 0 (0%) | 0 (0%) | 4 (4%) |
| Celastra-ceae | *Celastrus orbiculatus* | *Celastrus orbiculatus* | Species | Woody plant | L | 0 (0%) | 1 (2.9%) | 0 (0%) | 0 (0%) | 1 (1%) |
| Chenopo-diaceae | *Chenopodium album* | *Chenopodium album* | Species | Herbaceous plant | L | 0 (0%) | 0 (0%) | 1 (3.1%) | 0 (0%) | 1 (1%) |
| Com-melinaceae | *Commelina communis* var. *communis* | *Commelina communis* var. *communis* | Species | Herbaceous plant | L | 1 (4.5%) | 1 (2.9%) | 3 (9.4%) | 0 (0%) | 5 (5%) |
| Com-melinaceae | *Tradescantia fluminensis* | *Tradescantia fluminensis* | Species | Herbaceous plant | L | 0 (0%) | 1 (2.9%) | 0 (0%) | 0 (0%) | 1 (1%) |
| Convolvu-laceae | *Ipomoea* spp.-1 | *Ipomoea hederacea, Ipomoea nil* | Genus | Herbaceous plant | L | 0 (0%) | 0 (0%) | 1 (3.1%) | 0 (0%) | 1 (1%) |
| Convolvu-laceae | *Ipomoea* spp.-2 | *Ipomoea coccinea, Ipomoea quamoclit* | Genus | Herbaceous plant | L | 6 (27.3%) | 4 (11.4%) | 2 (6.3%) | 0 (0%) | 12 (12%) |
| Cupressa-ceae | *Metasequoia glyptostroboides* | *Metasequoia glyptostroboides* | Species | Woody plant | L | 7 (31.8%) | 2 (5.7%) | 1 (3.1%) | 0 (0%) | 10 (10%) |
| Cupressa-ceae | *Cryptomeria japonica* | *Cryptomeria japonica* | Species | Woody plant | L | 2 (9.1%) | 0 (0%) | 0 (0%) | 0 (0%) | 2 (2%) |

*(Continued)*

**Table 1.** (Continued)

| Family name | Identified plant taxa | Identified plant names | identified taxonomic level | Woody plant or Herbaceous plant | Database [1] | No. of fecal samples | | | | |
|---|---|---|---|---|---|---|---|---|---|---|
| | | | | | | Season | | | | all (n=100) |
| | | | | | | Spring (Mar. to May) (n=22) | Summer (Jun. to Aug.) (n=35) | Autumn (Sept. to Nov.) (n=32) | Winter (Dec. to Feb.) (n=11) | |
| Cupressaceae | *Juniperus chinensis* | *Juniperus chinensis* | Species | Woody plant | L | 0 (0%) | 0 (0%) | 1 (3.1%) | 0 (0%) | 1 (1%) |
| Daphniphyllaceae | *Daphniphyllum* spp. | *Daphniphyllum macropodum*, *Daphniphyllum teijsmannii* var. *teijsmannii* | Genus | Woody plant | L | 6 (27.3%) | 0 (0%) | 3 (9.4%) | 0 (0%) | 9 (9%) |
| Dioscoreaceae | *Dioscorea* spp. | *Dioscorea japonica*, *Dioscorea polystachya*, *Dioscorea tenuipes* | Genus | Herbaceous plant | L | 0 (0%) | 1 (2.9%) | 0 (0%) | 0 (0%) | 1 (1%) |
| Elaeagnaceae | *Elaeagnus pungens* | *Elaeagnus pungens* | Species | Woody plant | L | 1 (4.5%) | 3 (8.6%) | 0 (0%) | 0 (0%) | 4 (4%) |
| Elaeagnaceae | *Elaeagnus umbellata* var. *umbellata* | *Elaeagnus umbellata* var. *umbellata* | Species | Woody plant | L | 0 (0%) | 1 (2.9%) | 0 (0%) | 0 (0%) | 1 (1%) |
| Elaeocarpaceae | *Elaeocarpus zollingeri* var. *zollingeri* | *Elaeocarpus zollingeri* var. *zollingeri* | Species | Woody plant | L | 6 (27.3%) | 5 (14.3%) | 2 (6.3%) | 1 (9.1%) | 14 (14%) |
| Ericaceae | *Pieris japonica* | *Pieris japonica* | Species | Woody plant | L | 0 (0%) | 1 (2.9%) | 0 (0%) | 0 (0%) | 1 (1%) |
| Euphorbiaceae | *Croton tiglium* | *Croton tiglium* | Species | Woody plant | N | 0 (0%) | 1 (2.9%) | 0 (0%) | 0 (0%) | 1 (1%) |
| Euphorbiaceae | *Ricinus communis* | *Ricinus communis* | Species | Herbaceous plant | L | 0 (0%) | 1 (2.9%) | 0 (0%) | 0 (0%) | 1 (1%) |
| Fabaceae | *Desmodium paniculatum* | *Desmodium paniculatum* | Species | Herbaceous plant | L | 4 (18.2%) | 0 (0%) | 5 (15.6%) | 0 (0%) | 9 (9%) |
| Fabaceae | *Wisteria floribunda* [2] | *Melia azedarach*, *Wisteria floribunda* | Species | Woody plant | L | 0 (0%) | 3 (8.6%) | 1 (3.1%) | 0 (0%) | 4 (4%) |
| Fabaceae | *Pueraria montana* var. *lobata* | *Pueraria montana* var. *lobata* | Species | Herbaceous plant | L | 0 (0%) | 1 (2.9%) | 2 (6.3%) | 0 (0%) | 3 (3%) |
| Fabaceae | Fabaceae | *Acacia confusa*, *Acacia dealbata*, *Albizia julibrissin* | Family | Woody plant | L | 0 (0%) | 0 (0%) | 2 (6.3%) | 0 (0%) | 2 (2%) |
| Fabaceae | *Robinia pseudoacacia* | *Robinia pseudoacacia* | Species | Woody plant | L | 1 (4.5%) | 1 (2.9%) | 0 (0%) | 0 (0%) | 2 (2%) |
| Fagaceae | Fagaceae-1 | *Lithocarpus edulis*, *Lithocarpus glaber*, *Quercus phillyraeoides*, *Quercus serrata*, *Quercus variabilis* | Family | Woody plant | L | 12 (54.5%) | 23 (65.7%) | 15 (46.9%) | 5 (45.5%) | 55 (55%) |
| Fagaceae | Fagaceae-2 | *Castanopsis sieboldii* subsp. *sieboldii*, *Quercus dentata*, *Quercus glauca*, *Quercus serrata* | Family | Woody plant | L | 2 (9.1%) | 1 (2.9%) | 3 (9.4%) | 0 (0%) | 6 (6%) |
| Fagaceae | *Quercus* spp. | *Quercus acuta*, *Quercus glauca*, *Quercus myrsinifolia*, *Quercus serrata* | Genus | Woody plant | L | 9 (40.9%) | 24 (68.6%) | 6 (18.8%) | 0 (0%) | 39 (39%) |
| Fagaceae | *Quercus myrsinifolia* | *Quercus myrsinifolia* | Species | Woody plant | L | 0 (0%) | 0 (0%) | 1 (3.1%) | 0 (0%) | 1 (1%) |
| Hamamelidaceae | *Distylium racemosum* | *Distylium racemosum* | Species | Woody plant | L | 3 (13.6%) | 0 (0%) | 1 (3.1%) | 0 (0%) | 4 (4%) |
| Hydrocharitaceae | *Hydrilla verticillata* | *Hydrilla verticillata* | Species | Herbaceous plant | L | 0 (0%) | 0 (0%) | 1 (3.1%) | 0 (0%) | 1 (1%) |
| Lardizabalaceae | *Akebia quinata* | *Akebia quinata* | Species | Woody plant | L | 4 (18.2%) | 9 (25.7%) | 15 (46.9%) | 1 (9.1%) | 29 (29%) |
| Lauraceae | *Cinnamomum* spp. | *Cinnamomum camphora*, *Cinnamomum insularimontanum* | Genus | Woody plant | L | 15 (68.2%) | 30 (85.7%) | 29 (90.6%) | 5 (45.5%) | 79 (79%) |

*(Continued)*

**Table 1.** (Continued)

| Family name | Identified plant taxa | Identified plant names | identified taxonomic level | Woody plant or Herbaceous plant | Database [1] | No. of fecal samples | | | | all (n=100) |
|---|---|---|---|---|---|---|---|---|---|---|
| | | | | | | Season | | | | |
| | | | | | | Spring (Mar. to May) (n=22) | Summer (Jun. to Aug.) (n=35) | Autumn (Sept. to Nov.) (n=32) | Winter (Dec. to Feb.) (n=11) | |
| Lauraceae | *Neolitsea sericea* | *Neolitsea sericea* | Species | Woody plant | L | 6 (27.3%) | 4 (11.4%) | 2 (6.3%) | 0 (0%) | 12 (12%) |
| Lauraceae | *Machilus thunbergii* | *Machilus thunbergii* | Species | Woody plant | L | 0 (0%) | 3 (8.6%) | 2 (6.3%) | 0 (0%) | 5 (5%) |
| Lauraceae | *Cinnamomum camphora* | *Cinnamomum camphora* | Species | Woody plant | L | 0 (0%) | 1 (2.9%) | 2 (6.3%) | 0 (0%) | 3 (3%) |
| Menispermaceae | *Cocculus trilobus* | *Cocculus trilobus* | Species | Woody plant | L | 1 (4.5%) | 0 (0%) | 1 (3.1%) | 0 (0%) | 2 (2%) |
| Moraceae | *Broussonetia* spp. | *Broussonetia kazinoki, Broussonetia monoica* | Genus | Woody plant | N | 1 (4.5%) | 4 (11.4%) | 7 (21.9%) | 0 (0%) | 12 (12%) |
| Nyssaceae | *Camptotheca acuminata* | *Camptotheca acuminata* | Species | Woody plant | N | 0 (0%) | 1 (2.9%) | 0 (0%) | 0 (0%) | 1 (1%) |
| Oleaceae | Oleaceae-1 | *Fraxinus sieboldiana, Forsythia suspensa, Ligustrum japonicum* var. *japonicum, Ligustrum lucidum, Ligustrum ovalifolium,* | Family | Woody plant | L | 1 (4.5%) | 4 (11.4%) | 1 (3.1%) | 0 (0%) | 6 (6%) |
| Oleaceae | Oleaceae-2 | *Ligustrum lucidum, Ligustrum japonicum* var. *japonicum, Ligustrum ovalifolium, Forsythia suspensa* | Family | Woody plant | L | 8 (36.4%) | 18 (51.4%) | 18 (56.3%) | 3 (27.3%) | 47 (47%) |
| Oxalidaceae | *Oxalis articulata* | *Oxalis articulata* | Species | Herbaceous plant | L | 0 (0%) | 1 (2.9%) | 0 (0%) | 0 (0%) | 1 (1%) |
| Pentaphylacaceae | *Eurya* spp. | *Eurya emarginata* var. *emarginata, Eurya japonica* var. *japonica* | Genus | Woody plant | L | 1 (4.5%) | 0 (0%) | 0 (0%) | 1 (9.1%) | 2 (2%) |
| Pentaphylacaceae | *Ternstroemia gymnanthera* | *Ternstroemia gymnanthera* | Species | Woody plant | L | 0 (0%) | 0 (0%) | 1 (3.1%) | 0 (0%) | 1 (1%) |
| Pinaceae | *Pinus* spp. | *Pinus densiflora, Pinus thunbergii* | Genus | Woody plant | L | 5 (22.7%) | 0 (0%) | 0 (0%) | 0 (0%) | 5 (5%) |
| Pittosporaceae | *Pittosporum tobira* | *Pittosporum tobira* | Species | Woody plant | L | 0 (0%) | 0 (0%) | 0 (0%) | 2 (18.2%) | 2 (2%) |
| Platanaceae | *Platanus occidentalis* | *Platanus occidentalis* | Species | Woody plant | L | 6 (27.3%) | 3 (8.6%) | 2 (6.3%) | 0 (0%) | 11 (11%) |
| Poaceae | Poaceae-1 | *Anthoxanthum odoratum, Bromus diandrus, Bromus japonicus, Bromus remotiflorus, Hordeum vulgare* subsp. *vulgare* | Family | Herbaceous plant | L | 0 (0%) | 0 (0%) | 1 (3.1%) | 0 (0%) | 1 (1%) |
| Poaceae | Poaceae-2 | *Andropogon virginicus, Calamagrostis epigeios, Coix lacryma-jobi, Miscanthus floridulus, Miscanthus sacchariflorus, Miscanthus sinensis, Hemarthria sibirica, Imperata cylindrica* var. *major, Sorghum halepense* | Family | Herbaceous plant | L | 1 (4.5%) | 1 (2.9%) | 1 (3.1%) | 0 (0%) | 3 (3%) |
| Poaceae | *Digitaria* spp. | *Digitaria ciliaris, Digitaria radicosa* | Genus | Herbaceous plant | L | 0 (0%) | 0 (0%) | 1 (3.1%) | 0 (0%) | 1 (1%) |
| Poaceae | *Elymus kamoji* | *Elymus kamoji* | Species | Herbaceous plant | L | 0 (0%) | 1 (2.9%) | 0 (0%) | 0 (0%) | 1 (1%) |
| Polygonaceae | *Persicaria* spp. [3] | *Hydrilla verticillata, Persicaria hydropiper, Persicaria longiseta, Persicaria maculosa* var. *pubescens, Persicaria posumbu* var. *laxiflora* | Genus | Herbaceous plant | L | 8 (36.4%) | 14 (40%) | 11 (34.4%) | 0 (0%) | 33 (33%) |

*(Continued)*

**Table 1.** (Continued)

| Family name | Identified plant taxa | Identified plant names | identified taxonomic level | Woody plant or Herbaceous plant | Database [1] | No. of fecal samples | | | | all (n=100) |
|---|---|---|---|---|---|---|---|---|---|---|
| | | | | | | Season | | | | |
| | | | | | | Spring (Mar. to May) (n=22) | Summer (Jun. to Aug.) (n=35) | Autumn (Sept. to Nov.) (n=32) | Winter (Dec. to Feb.) (n=11) | |
| Rosaceae | Rosaceae | *Chaenomeles speciosa, Eriobotrya japonica, Photinia glabra, Photinia villosa, Rhaphiolepis umbellata, Pyracantha angustifolia, Pyrus calleryana, Sorbus alnifolia* | Family | Woody plant | L | 10 (45.5%) | 19 (54.3%) | 20 (62.5%) | 4 (36.4%) | 53 (53%) |
| Rosaceae | *Rubus* spp. | *Rubus hirsutus, Rubus parvifolius, Rubus sumatranus* | Genus | Woody plant | L | 1 (4.5%) | 14 (40%) | 6 (18.8%) | 0 (0%) | 21 (21%) |
| Rosaceae | *Prunus serrulata* var. *spontanea* | *Prunus serrulata* var. *spontanea* | Species | Woody plant | L | 4 (18.2%) | 2 (5.7%) | 5 (15.6%) | 0 (0%) | 11 (11%) |
| Rosaceae | *Rubus buergeri* | *Rubus buergeri* | Species | Woody plant | L | 0 (0%) | 1 (2.9%) | 0 (0%) | 0 (0%) | 1 (1%) |
| Rubiaceae | *Paederia foetida* | *Paederia foetida* | Species | Herbaceous plant | L | 0 (0%) | 0 (0%) | 1 (3.1%) | 1 (9.1%) | 2 (2%) |
| Sapindaceae | *Acer* spp. | *Acer buergerianum, Acer palmatum* | Genus | Woody plant | L | 8 (36.4%) | 17 (48.6%) | 4 (12.5%) | 2 (18.2%) | 31 (31%) |
| Solanaceae | *Solanum lyratum* var. *lyratum* | *Solanum lyratum* var. *lyratum* | Species | Herbaceous plant | L | 7 (31.8%) | 7 (20%) | 4 (12.5%) | 0 (0%) | 18 (18%) |
| Theaceae | *Camellia* spp. | *Camellia japonica, Camellia sasanqua, Camellia sinensis* | Genus | Woody plant | L | 1 (4.5%) | 1 (2.9%) | 2 (6.3%) | 0 (0%) | 4 (4%) |
| Ulmaceae | *Zelkova serrata* | *Zelkova serrata* | Species | Woody plant | L | 0 (0%) | 0 (0%) | 1 (3.1%) | 0 (0%) | 1 (1%) |

using the Bray-Curtis and Jaccard indices. The NMDS plot did not indicate seasonal differences in diet (Fig 3), nor did the results of the PerMANOVA (P = 0.11, F = 1.3, R2 =0.04 using the Bray-Curtis index, P = 0.08, F = 1.2, R2 =0.04 using the Jaccard index) (see S2 Fig, the results of the analysis without removing low-frequency ASVs). The top four food plant taxa in all seasons were *Cinnamomum* spp., Fagaceae-1, Rosaceae, and Oleaceae-2, indicating that the main food plant resources in this study area remained consistent throughout the year (Table 1). Coverage-based rarefaction and extrapolation curves were constructed and compare food plant diversity across seasons (Fig 4). The results showed that spring, summer, and autumn samples provided approximately 90% sample coverage, whereas winter samples had lower coverage at around 80%. When comparing food plant diversity based on winter sample coverage, the 95% confidence intervals did not overlap with those of the other seasons, indicating lower food plant diversity in winter. The average number of identified plant taxa per fecal sample was 6.8 (median: 6, min: 1, max: 16). Although the number of winter fecal samples in this study was limited (Table 1), a comparison of the number of food plant taxa in a single fecal sample across the four seasons, significantly fewer plant taxa were found in feces sampled in winter than in other seasons (Steel-Dwass test, P<0.05) (Fig 5).

1) Database used for homology search (L: *rbcL* Local database, N: NCBI). For more details, see S5 Table.

2) The sequence of *Melia azedarach* (HE963559) was analyzed using homology search in NCBI, and misidentification was confirmed using Blast Tree View. The results showed that *M. azedarach* (HE963559) clustered with *Wisteria* spp. Therefore, the sequence of *M. azedarach* (HE963559) was considered to be misidentified and were excluded from this study.

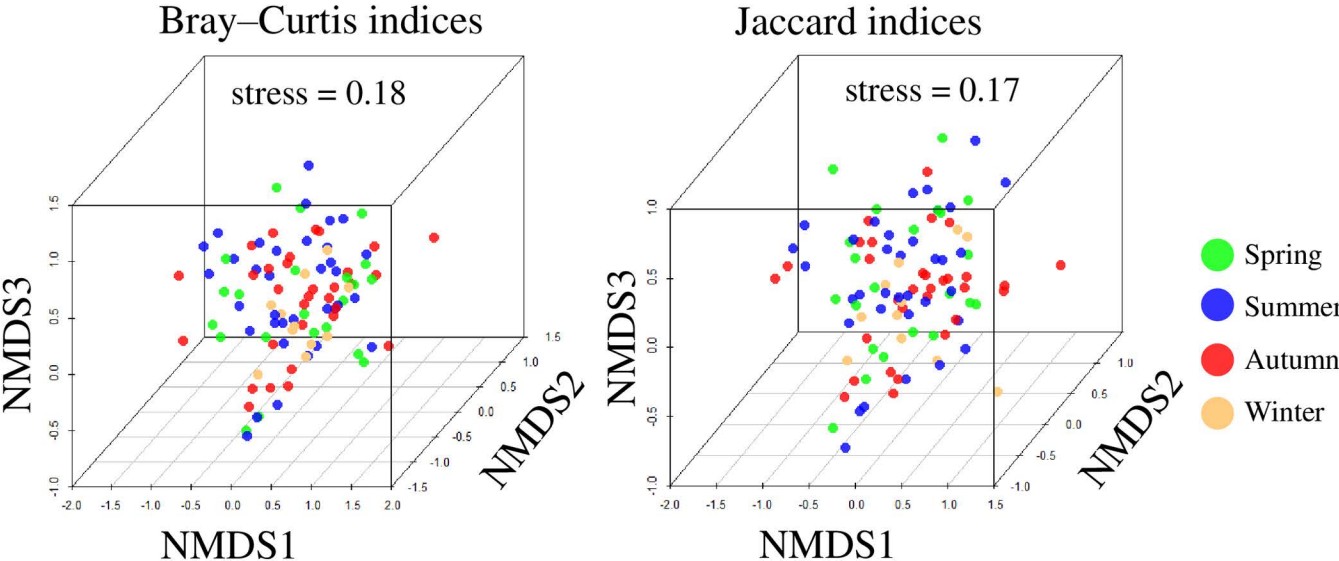

**Fig 3. Non-metric multidimensional scaling (NMDS) plot based on Bray–Curtis and Jaccard indices obtained from *rbcL* analyses.** To visualize both the Bray–Curtis and Jaccard indices, one outlier was removed from each dataset.

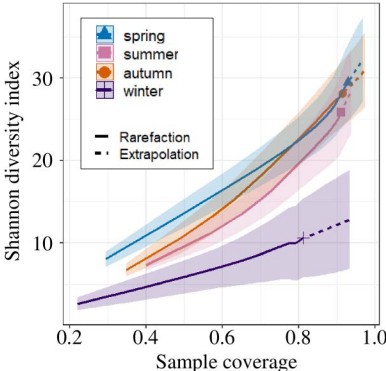

**Fig 4. Coverage-based rarefaction and extrapolation curve.** Shaded areas represent 95% confidence intervals obtained using a bootstrap method with 1,000 replicates.

3) The sequences of *Hydrilla verticillata* (KJ747463–KJ747466) were analyzed using a homology search in NCBI, and misidentification was confirmed using Blast Tree View. The results showed that *H. verticillata* (KJ747463–KJ747466) clustered with Polygonaceae. Therefore, these sequences were considered to be misidentified and were excluded from this study.

## Discussion

### Accuracy of Food plant resource identification

The main objectives of this study were to elucidate the dominant food plant resources of *A. speciosus* inhabiting artificial green spaces and to determine whether constructing a local database using NCBI data improves classification resolution. In this study, a total of 72 plant

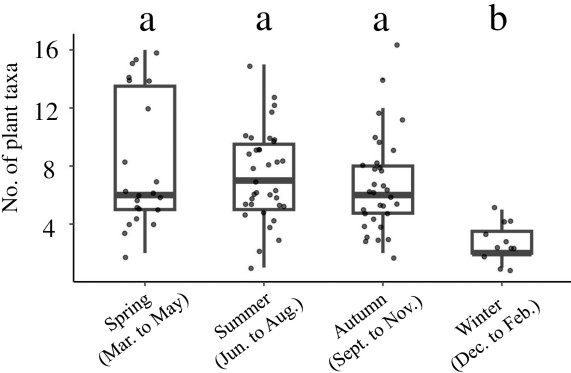

**Fig 5. Comparison of the number of plant taxa per fecal sample identified in different seasons.** The box for each season represents the interquartile range (25th to the 75th percentile), with the median number of taxa for each season indicated by a bold line within the box. The 'whiskers' extend to the most extreme values within 1.5 times the interquartile range from the median. Statistically significant differences between the four seasons, as determined by the Steel-Dwass test ($P < 0.05$), are indicated by different letters.

taxa were identified from 100 fecal samples, with 43 (59.7%) assigned to species, 16 (22.2%) to genus, and 13 (18.1%) to family. In previous studies, the food plant resources utilized by *A. speciosus* in Hokkaido Island in northern Japan [37], and in the Seto Inland Sea of Western Japan [37,38] were identified by HTS using partial sequences of the chloroplast *trnL* P6 loop intron region without constructing a local database [37–39]. Although the survey site, survey period (Sato et al. [37] is June, July, August, September, October; Sato et al. [38] is February, March, April, May; Sato et al. [39] is March, April, June, July, August), and DNA region sequenced by HTS differed between this study and the aforementioned reports [37–39], the resolution of plant species identification in this study was higher than those in the previous studies. One possible reason for this may be because the previous studies did not construct a local database of the *trnL* P6 loop intron region for their respective study areas. Another possible reason could be differences in the classification resolution characteristics of the analyzed DNA regions; for example, as the classification resolution superiority of the rbcL region and the trnL P6 loop varied across reports [40,43,64], rbcL may have been more suitable for the *A. speciosus*. Alternatively, it could be due to a combination of both of these factors. In addition, the number of plant taxa detected per fecal sample in this study was higher than that in these previous studies [37,38]. Since the number of fecal samples analyzed in Innoshima Island in the Seto Inland Sea of western Japan, is small [39]; therefore, it was not included in the comparison. In this study, ca. 60 mg dry weight of the feces excreted by a single trapped individual were analyzed, whereas in the previous studies [37,38], only three fecal pellets (weight not reported, but likely less than 60 mg dry weight) collected from one trapped individual were analyzed; this may have led to an underestimation of the food resources foraged by one individual. To the best of our knowledge, no DNA metabarcoding reports have been published on the optimal amount of feces required for elucidating the diet of small mammals such as *A. speciosus*. To establish a baseline for feces-based diet estimation, it may be necessary in the future to verify the optimal amount of fecal material required for such analyses, as well as to assess the effectiveness of different methods for homogenizing fecal samples.

## Dominant food plant resources

In this study, the dominant families in the fecal samples were Lauraceae (81.0%) (mostly *Cinnamomum* spp.), followed by Fagaceae (70.0%) (mostly *Quercus* spp.), Rosaceae (68.0%),

and Oleaceae (48.0%). The results are consistent with previous reports which showed that *A. speciosus* utilizes a variety of food plants [34,35,37,39,65,67]. A previous study showed that the diet of *A. speciosus* changes significantly depending on the season [37,39]. However, the findings of this study showed the food plant resources utilized by *A. speciosus* did not differ among four seasons (Fig 3). The dominant food plant resources were the same across all seasons. Although the number of winter fecal samples in this study was limited, in winter, both diet diversity and the number of plant taxa identified per single sample in winter were significantly lower than that in other seasons (Figs 4 and 5). These results may reflect the fact that the vegetation at the study site in this research consists mainly of evergreen broad-leaf forest, in contrast to the study site in Hokkaido Island [37], which is deciduous broad-leaf forest, and the study site in the Seto Inland [38,39], which is mixed forest dominated by broad-leaf forest, including evergreen broad-leaf forest and evergreen needle-leaf forest (S3 Fig) [68]. The foraging characteristics observed in this study area suggest that the artificial vegetation environment may be unfavorable for *A. speciosus*, potentially limiting their range of food choices.

In the Japanese Archipelago, the diet of *A. speciosus* is generally comprised of the nuts of members of the Fagaceae, such as *Quercus* sp. [65]. These findings have also been corroborated by DNA metabarcoding [37–39]. However, in this study, materials from the family Lauraceae (81.0% of all 100 fecal samples) were dominant in fecal samples, with *Cinnamomum* spp. (79.0%) (presumed to be either *C. camphora* or *C. insularimontanum* based on our vegetation survey) also identified as a dominant food plant resource for the *A. speciosus* population at this study area throughout the year. In our study area, members of the Lauraceae, which had the highest foraging rate and was utilized throughout the year, were not detected in the diet of *A. speciosus* on Hokkaido Island in northern Japan [37], and only in the summer on Innoshima Island in the Seto Inland Sea of western Japan [39]. While there have been reports of *Cinnamomum* spp. being utilized as a food resource by *A. speciosus*, to the best of our knowledge, there are no previous records of this genus being a dominant food plant source for this mouse [34,35,37,39,65–67].

*Cinnamomum* spp., the most dominant species in our study area, are evergreen trees. *Cinnamomum camphora* flowers, about 5 mm in diameter, bloom from May to June and bear green fruits measuring approximately 8 mm in diameter in late summer, which eventually turn black upon ripening in autumn. On the other hand, the flowering and fruiting characters of *C. insularimontanum* are similar to those of *C. camphora*; they have flowers approximately 5 mm in diameter, bloom from June to July and bear green fruits measuring approximately 11–13 mm in diameter in late summer, which subsequently turn black upon ripening in autumn. Given that these *Cinnamomum* spp. retain their foliage throughout the year [69]. Therefore, it is believed that *A. speciosus* forages on the leaves, fruits and seeds of *Cinnamomum* spp. throughout the year. As mentioned previously in the discussion above, *A. speciosus* is known to prefer nuts from members of the Fagaceae, such as *Quercus* sp. [37,39,65]. However, based on the results of our vegetation survey, there were almost no *Quercus* trees with acorns in the study area, suggesting that leaves, fruits and seeds of *Cinnamomum* spp., Rosaceae, Oleaceae-2 might be utilized during the winter. The herbaceous plants that were utilized by *A. speciosus* at the study site were dominated by *Persicaria* sp. (33.0%), followed by *Solanum lyratum* var. *lyratum* (18.0%) and members of the Asteraceae (15.0%) (Table 1). The *A. speciosus* population at the study area fed more on woody plants than on herbaceous plants, which is consistent with the conclusions of previous studies [37–39].

The planting and maintenance of woody plants is therefore considered to be important for maintaining the population of *A. speciosus* in the study area. In addition, in order to develop artificial green spaces in the study area in a way that considers the habitat suitability of *A. speciosus*, it is considered necessary to plant and maintain forage plants according to the season.

## Limitations of this study

This study has several limitations that affected estimates of diet composition by DNA metabarcoding.

The first problem was the absence of a negative control in the HTS experiment to confirm sequences derived from contamination within the laboratory [70,71]. As a result, sequences of Japanese alpine plants (*G. nipponica*, *S. pentapetala*, and *Vaccinium* sp.), which do not grow in the study area, were detected in the feces of *A. speciosus* (S5 Table). These alpine plants may be due to contamination by material that was being analyzed at the same time in the laboratory. In this study, since we were able to refer to a detailed plant list of the study area, we could remove the *rbcL* F3R3 sequences of plants not growing in the study area. In plant resource analyses employing highly sensitive HTS methods, DNA contamination from other organisms in the laboratory can also be a problem. Consequently, methods such as using negative controls to check for contamination or creating a detailed plant list of the study area are considered necessary. However, plant contamination from pollen during the field sampling process or from secondary predation of insects consumed by *A. speciosus* [72] cannot be removed. The presence of these factors must always be taken into account when interpreting the results. Since the objective of this study was to analyze the dominant plant species utilized by the *A. speciosus* population, low-frequency ASVs were removed using a 1.0% threshold. However, this threshold and procedure are also subject to debate [73,70].

The second problem was that the number of entries in the NCBI database is still insufficient. According to the environmental assessment conducted prior to this study, 796 plant species grow at the study area. However, we were unable to determine the *rbcL* F3R3 sequences for all of the plants in this study area, and some plant species did not have *rbcL* F3R3 sequences registered in the NCBI database (S1 Table). Fortunately, most of the plant species identified as plant food resources in the study had *rbcL* F3R3 sequences that were registered in the NCBI database, so we were able to construct a local database that could be used to accurately identify food resources. However, it is not always possible to construct a local database covering all of the plants growing at a study area. To facilitate the construction of local databases for different research regions using this method, it is essential to continue expanding the data available in the NCBI database.

The final limitation of the study was the selection of capture bait. In this study, the *rbcL* F3R3 sequences of the walnuts (*J. regia*) used as bait and those of *P. rhoifolia* and *P. stenoptera* which grow in the study area were the same, making it impossible to distinguish between them. When small mammals, such as *A. speciosus*, are captured using bait traps to estimate food plant resources in their feces, care should be taken to use bait derived from plants that are not genetically related to the plant species at the study site.

## Conclusions

In conclusion, the DNA metabarcoding analysis described in this study improves our understanding of the feeding habits of *A. speciosus* inhabiting artificial green spaces. In this study, in order to construct the local *rbcL* F3R3 database, the *rbcL* F3R3 sequences for plant species that could not be collected in the study area were obtained from the NCBI database. As a result, we were able to identify the food plant resources of *A. speciosus* at a resolution higher than those reported in previous studies employing DNA metabarcoding. This is the first study to thoroughly elucidate the food plant resources of the *A. speciosus* living in artificial green spaces.

## Supporting information

**S1 Table. The vegetation at the study site and constructed *rbcL* local database.**
(XLSX)

**S2 Table. List of initial PCR primers.** List of primers used for the initial PCR. Letters in italics indicate the MiSeq sequencing primers. Bold Ns indicate random bases used to improve the quality of MiSeq sequencing. Single underlined letters indicate DNA barcoding primer sequences.
(XLSX)

**S3 Table. List of second PCR primers.** List of primers used for the second PCR. Letters in italics indicate the MiSeq sequencing primers. Bold Xs indicate index sequences used to identify each sample. Double-underlined characters indicate P5/P7 adapter sequences for MiSeq sequencing.
(XLSX)

**S4 Table. Sequencing statistics of *rbcL* regions.** Input reads: number of reads in raw FASTQ files. Filtered reads: number of reads after preliminary quality filtering. Denoised F and R reads: number of reads after quality filtering. Merged reads: number of merged forward-reverse reads. Nonchim reads: number of merged reads after removal of chimeric sequences. Low frequency removal reads: low-frequency ASVs, defined as those representing fewer than 1.0% of the total number of sequences in each fecal sample.
(XLSX)

**S5 Table. Results of homology searches for fecal samples using the local *rbcL* database and the NCBI database.**
(XLSX)

**S1 Fig. The result of rarefaction curve constructed using ASV data after the removal of low-frequency ASVs (1.0%).**
(PDF)

**S2 Fig. Non-metric multidimensional scaling (NMDS) plot based on Bray–Curtis and Jaccard indices using all ASV information without removing low-frequency ASVs.** The NMDS plot did not indicate seasonal differences in diet, nor did the results of the PerMANOVA (P = 0.15, F = 1.3, R2 =0.04 using the Bray-Curtis index, P = 0.07, F = 1.2, R2 =0.04 using the Jaccard index). These results were similar to the NMDS plot (Fig 3) and PerMANOVA analysis results after removing low-frequency ASVs.
(PDF)

**S3 Fig. The Land-Cover Map of the area surrounding the in the previous studies and this study of the food plant resources utilized by the Large Japanese field mice (*Apodemus speciosus*) using DNA metabarcoding.** These maps were draw used JAXA High Resolution Land-Use and Land-Cover Map of Japan [68]. (A) ▲ indicate Forest Research Station, the Field Science Center for Northern Biosphere, Hokkaido University. (B) Sato et al. 2019 [38] was conducted survey on Ikuchijima, Hakatajima, Kamikamagarijima, Ohmishima, Ohsak-ishimojima, Ohshima and Shimokamagarijima. Sato et al. 2022 [39] was conducted survey on Innoshima. (C) Dotted lines indicate areas where the Large Japanese field mice (*Apodemus speciosus*) were trapped. Republished from Homepage of High-Resolution Land Use and Land Cover Map Products (https://earth.jaxa.jp/en/data/2562/index.html) under a CC BY license, with permission from Japan Aerospace Exploration Agency, original copyright 3 Feb. 2025.
(DOCX)

## Acknowledgements

We thank the staff of the Aichi Complex of Idemitsu Kosan Co., Ltd., Chita Thermal Power Station of JERA Co., Inc. (formerly Chita Thermal Power Station of Chubu Electric Power Co., Ltd.), and ENEOS Corporation for access to the study area. We thank Mr. Nobuyuki Kitou and Ms. Akane Shibata of Chubu University for their assistance with capturing mice.

## Author contributions

**Conceptualization:** Motoyasu Minami.

**Data curation:** Hirokazu Kawamoto.

**Formal analysis:** Taichi Fujii.

**Funding acquisition:** Motoyasu Minami.

**Investigation:** Taichi Fujii, Hirokazu Kawamoto, Tomoyasu Shirako, Motoyasu Minami.

**Methodology:** Tomoyasu Shirako, Masatoshi Nakamura.

**Project administration:** Motoyasu Minami.

**Writing – original draft:** Taichi Fujii.

**Writing – review & editing:** Tomoyasu Shirako, Masatoshi Nakamura, Motoyasu Minami.

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
