## [Decision Letter · Decision Letter 0]

9 Jun 2024

PONE-D-24-12099Identification of Large Japanese field mouse *Apodemus speciosus* food plant resources in an industrial green space using DNA metabarcodingPLOS ONE

Dear Dr. Fujii,

Thank you for submitting your manuscript to PLOS ONE. After careful consideration, we feel that it has merit but does not fully meet PLOS ONE’s publication criteria as it currently stands. Therefore, we invite you to submit a revised version of the manuscript that addresses the points raised during the review process.

**ACADEMIC EDITOR:** Be sure to:

For your submission "Identification of Large Japanese field mouse Apodemus speciosus food plant resources in an industrial green space using DNA metabarcoding" to PLOS ONE to proceed further in the review process, you must make major revisions to your manuscript. Revise it as suggested by the reviewer.

We look forward to receiving your revised manuscript.

Kind regards,

Faham Khamesipour, Ph.D.

Academic Editor

PLOS ONE

Journal Requirements:

2. We note that [Figure 1] in your submission contain [map/satellite] images which may be copyrighted. All PLOS content is published under the Creative Commons Attribution License (CC BY 4.0), which means that the manuscript, images, and Supporting Information files will be freely available online, and any third party is permitted to access, download, copy, distribute, and use these materials in any way, even commercially, with proper attribution. For these reasons, we cannot publish previously copyrighted maps or satellite images created using proprietary data, such as Google software (Google Maps, Street View, and Earth). For more information, see our copyright guidelines: http://journals.plos.org/plosone/s/licenses-and-copyright.

Additional Editor Comments:

For your submission "Identification of Large Japanese field mouse Apodemus speciosus food plant resources in an industrial green space using DNA metabarcoding" to PLOS ONE to proceed further in the review process, you will need to make major revisions to your manuscript. Revise it as suggested by worty reviewer.

Reviewers' comments:

Reviewer's Responses to Questions

**Comments to the Author**

1. Is the manuscript technically sound, and do the data support the conclusions?

Reviewer #1: Partly

Reviewer #2: Partly

2. Has the statistical analysis been performed appropriately and rigorously?

Reviewer #1: Yes

Reviewer #2: No

3. Have the authors made all data underlying the findings in their manuscript fully available?

Reviewer #1: Yes

Reviewer #2: Yes

4. Is the manuscript presented in an intelligible fashion and written in standard English?

Reviewer #1: Yes

Reviewer #2: Yes

5. Review Comments to the Author

Reviewer #1: This study examined the plant diets of the large Japanese field mouse Apodemus speciosus (endemic to Japan) inhabiting an artificial green space using the DNA metabarcoding method. This method is very useful method because it could clarify the multiple diet simultaneously from feces or gastrointestinal tract using the 2nd generation sequencing technology and therefore has been extensively applied to various studies to understand the dietary trend of world mammals. Japan is no exception. There are five dietary studies using this method even just showing the analyses for rodents in Japan (Sato et al. 2018 Journal of Mammalogy, Sato et al. 2019 Mammal Study, Sato et al. 2022 Mammal Research, Sato et al. 2023 Mammal Study, and Murano et al. 2023 Mammal Study). This study is sixth paper that would contribute to these progresses. Plant dietary analyses for Apodemus speciosus have already been conducted in Sato et al. (2018) in Hokkaido (northernmost islands in Japan) and Sato et al. (2019, 2022) in Seto Inland Sea islands (southwestern parts of Japan). This study conducted in a central part of Japan is the third study to examine the plant diet of A. speciosus.

On the other hand, it is new to see that this study examined the dietary trend of rodents in an artificial green space in urban environment. As far as I know, there are 27 papers for the DNA metabarcoding analyses of rodent species worldwide that have never provided this perspective of diet on the artificial space. Furthermore, this study seriously considered local DNA database for species identification and provided higher resolution for the dietary species of A. speciosus. In these regards, this study is worthwhile to be published. However, I need to understand this paper more carefully by resolving issues listed below before recommending publication. Furthermore, some important papers as listed above or below are not cited properly. I cannot recommend it for publication without proper resolution of these issues. See below for my major and minor concerns.

Major concerns

1. L46-60: The most important paragraph in introduction showing the motivation of this study is the 1st paragraph. However, almost all of references (9/12) are from papers written in Japanese. In this situation, it is difficult to convince worldwide researchers to understand that the artificial green space that can be regarded as a space used in OECM is important place. Cite more worldwide papers. If this idea began in Japan, please explain so and show that almost nothing has been conducted in the world except for Japan.

2. Didn’t you examine negative control analyses, for example, at the DNA extraction or PCR steps? Such negative control PCR samples should be sequenced and treated appropriately in the sequence read matrix.

3. This study is important because it provides information of diets of A. speciosus in an artificial environment in urban place and central part of Japan. However, there are no discussions in these points. How different are the diets of field mice in northern (Hokkaido), central (Aichi), and southern (Seto Inland Sea) parts of Japan? How different are the diets of field mice between urban and natural environments? These discussions should be made.

Minor concerns

L64: inhabited > inhabits [this species is also inhabiting forest now]

L69: I am not sure if only conserving preferred resources are good or not in terms of ecosystem conservation. It may be good for A. speciosus, but it must be considered that the ecosystem consists of diverse organisms. The idea shown here is too simplistic.

L72-73: The sentence may not be accurate. There are several traditional studies using the morphological analyses of the stomach contents, which were conducted on the forest and around the cultivated field (considered artificial green space in the Satoyama [OECM] landscape). These studies also examined food plant resources of A. speciosus. So, you should cite these traditional papers if you would like to express so like the current sentence. Papers that you cited are a DNA barcoding study using subcloning method [17] and DNA metabarcoding studies [18-19]. In this case, you should add Sato et al. (2022: Mammal Research 67: 109–122) here.

L74-77: All references are not from studies of A. speciosus. Because you started this paragraph from the story of A. speciosus, you should introduce such traditional studies of A. speciosus.

L79: [20] is not a study of A. speciosus.

L81: You should add Sato et al. (2022: Mammal Research 67: 109–122) here. In addition, [27-38] is not appropriately selected but seems to be selected randomly. You should cite only relevant papers, for example, papers on DNA metabarcoding method applied to rodents. Several reviews can be cited here, e.g.,

Alberdi et al. (2018) Methods in Ecology and Evolution 9: 134-147

Alberdi et al. (2019) Molecular Ecology Resources 19: 327–348.

Ando et al. (2020) Environmental DNA 2: 391–406. [already cited]

Bohmann et al. (2022) Molecular Ecology Resources 22: 1231–1246.

Tedersoo et al. (2022) Molecular Ecology 31: 2769–2795.

Tercel et al. (2021) Molecular Ecology 30: 2199–2206.

L83: You should add Sato et al. (2022: Mammal Research 67: 109–122) here.

L83-86: You may misunderstand between low resolution and low accuracy. Both studies showed accurate identifications but trnL mostly showed family level identification (low resolution) without a local plant list or DNA database. Change the expression.

L87: Also consider the usage of accuracy.

L89: [19] and Sato et al. (2022) above also used this marker.

L91: Just long is not enough for the reason to use this marker. Previously 7 of 27 papers of rodent dietary DNA metabarcoding studies used ITS2 marker (e.g., Murano et al. 2023 and Sato et al. 2023). Only two studies used rbcL marker (Latinne et al. 2014 Journal of Cave and Karst Studies 76: 139–145, Gabrielson et al. 2023 Biological Invasions, https://doi.org/10.1007/s10530-023-03159-4). ITS2 is longer than rbcL and probably more informative, but it may show less taxonomic coverage than rbcL. So, you must not oversimplify the background information. You should select markers in terms of length, informativeness (genetic variation), taxonomic coverage, or cost (long marker requires expensive sequencing kit), not just based on length.

L131-132: What is the procedure “dried overnight at 60 degree” for? It seems to lead to DNA degradation. What do you think?

L141, 151: Write a full name of the kit KAPA HiFi.

L147-148: Did you perform “ligation” to put both indices and adaptors or put them through 2nd PCR? I suppose that the latter is true, but your explanation looks the opposite.

L195: Consider the usage of accurate.

L224, 231: Consider the usage of accuracy.

L251-255: Why were these plants excluded. At least genus level discussion can be made. Even if these species grow near the capture site, contamination could not necessarily occur. What is the mechanism for the contamination. Pollen? The reason for the exclusion is not clear.

L336, 354, 360: Consider the usage of accuracy.

L360-362: You used rbcL for its longer sequence (as noted in Introduction), but there are no discussions regarding this length (variation) issue. Also discuss informativeness of these markers. Because ITS2 has been the most frequently used markers for plant dietary species, you should also discuss the usefulness of rbcL compared with the ITS2 (UniPlant marker).

L369-372: See Sato et al. (2022) in Mammal Research above and discuss with it regarding Lauraceae.

L373: You should add Sato et al. (2022: Mammal Research 67: 109–122) here.

L379: You assumed “seeds”, but looking at Fig. 3, Fagaceae or Quercus is not so abundant in autumn. Is it correct to say that these are from acorns?

L389-402: Sato et al. (2018) and Sato et al. (2022) showed seasonal variations of the diet of A. speciosus in Hokkaido and Seto Inland Sea, respectively. Similar trend in the diet of this species among seasons is not usual for this species. You should discuss this point and assess if this unusual trend was caused by an artificial environment or not.

L401-402: Why did you think Quercus is needed in the study place? This species could live in the place for more than 50 years. This suggestion has no rationale.

L464: Iwase > Iwasa

L480: Apodemus should be italic.

Reviewer #2: The authors used DNA metabarcoding to investigate dietary resource use of a common, but endemic rodent species in an artificial green space. Studying the diet of the primary consumers in these spaces can be important for facilitating a resilient food web within cityscapes. I applaud the authors for good writing, including their analyzed datasets as supplementary data, and for thoroughly developing (and validating) a local reference database to increase certainty on the context of the molecular data. Along those lines, I appreciate the efforts of DNA barcoding over 100 new plants and depositing them in local databases. I was particularly fascinated by the seasonal variation found in this study.

Along with these merits, there are some issues I challenge the authors to address. There is a need for more precise language regarding the interpretation (does all metabarcoding data equal diet?), and a need for more statistical support, particularly for any interpretations of seasonal variation. Similarly, there were some key results that were missing methods. I am also concerned that some of the pre-processing steps led to extraneous data loss.

Finally, I recommend elevating the discussion to focus more on the feeding ecology of the animal with respect to the artificial setting vs a natural setting. The discussion largely repeats the results section with little “why”. Consider drawing from the resource/habitat selection literature (e.g., optimal foraging/diet theory). The authors should scrutinize the taxa and consider their nutritional pros or cons, phenology, and possibly competitors in the area. More minor, I also recommend including common names with the taxonomic data.

I hope the authors find the following publications helpful regarding dietary interprations from metabarcoding data:

Nielsen JM, Clare EL, Hayden B, Brett MT, Kratina P. 2018. Diet tracing in ecology: method comparison and selection. Methods in Ecology and Evolution. 9(2):278–291. doi:10.1111/2041-210X.12869.

Clare EL. 2014. Molecular detection of trophic interactions: emerging trends, distinct advantages, significant considerations and conservation applications. Evolutionary Applications. 7(9):1144–1157. doi:10.1111/eva.12225.

Introduction:

Line 73: Be careful with the word “preference”. We can determine what an animal uses or selects within particular spatial and temporal scales but can we really know what an animal prefers? I recommend the reference below.

Thomas DL, Taylor EJ. 1990. Study Designs and Tests for Comparing Resource Use and Availability. The Journal of Wildlife Management. 54(2):322–330. doi:10.2307/3809050.

Lines 68-86: This content must better describe what we know about the animal’s phenology and diet, for example, in undisturbed or less disturbed habitat. Do they consume foliage, seeds, roots … etc.? Are they strict herbivores? Are they seed predators, do they disperse seeds? Most rodents are omnivores in order Rodentia. If A. speciosus is an omnivore, I suggest better justifying why you only investigate herbivory and not other trophic levels. If the animal consumes insects, can they assist with pest control, for example? Could any of the plants detected in the feces actually be plants that an insect consumed? Do they hibernate? Do they cache seeds for the winter? Overall, this content should be much improved by delving deeper into the study system and what the species would need if placed in an environment of human design.

Lines 87-93: It seems like the authors present a false dichotomy between trnL and rbcL. Multiple markers are better than one and barcodes other than trnL and rbcL are available for plant metabarcoding, such as ITS2. ITS2 is also commonly used and delivers even better taxonomic resolution than rbcL. My main recommendation would be to also include ITS2 in the dataset, if funding permits; however, I would at least mention ITS2 and make a better case for only using rbcL. I hope the authors find the two publications below helpful.

Fahner NA, Shokralla S, Baird DJ, Hajibabaei M. 2016. Large-Scale Monitoring of Plants through Environmental DNA Metabarcoding of Soil: Recovery, Resolution, and Annotation of Four DNA Markers. PLOS ONE. 11(6):e0157505. doi:10.1371/journal.pone.0157505.

Thongjued K, Garcia K, Scott D, Gonthier DJ, Dupuis JR. DNA metabarcoding diet analysis in a generalist omnivore: feeding trials reveal the efficacy of extraction kits and a multi-locus approach for identifying diverse diets. Integrative Zoology. n/a(n/a). doi:10.1111/1749-4877.12806. https://onlinelibrary.wiley.com/doi/abs/10.1111/1749-4877.12806.

Lines 96-98. What does “better” mean? Metabarcoding is only one dimension of the diet. That is, we can only detect what is being egested. “Better” can also mean what the animals are metabolizing and incorporating, or even the fitness benefits a dietary resource confers. These final sentences also require a hypothesis and predictions, seeing as there are some diversity analyses and statistics for them. Set the stage for our expectations as they relate to the biology of the animal.

Methods:

Line 115: Use “We live-captured at total of…”. Instead of “Collection”. “Collection” may imply to some readers that animals were captured and killed for museum collection.

Lines 121-123: Describe the sterility of the trap sampling procedures. Contamination in the field is difficult to prevent. We could also use more information on trapping logistics. Were traps set along a transect? Or were they opportunistically placed?

I am concerned that there is no mention of DNA extraction controls, PCR-negative controls, or positive controls? Were these not included in the laboratory work? If they were, how many? These controls should always be sequenced along with field-collected samples in metabarcoding studies.

Lines 131-134: Citations/justification needed for the pre-incubation step and DNA extraction kit. I’m personally curious but perhaps others will be too.

Lines 136-137: Refer specifically to Illumina sequencing technology (e.g., Illumina). Not all HTS technologies are sensitive to over-clustering because they can have different chemistries (e.g., Oxford Nanopore).

Line 140: Cite the frame-shifting primer method.

Lines 142 and 151-152: Only report final concentrations of the primers, not the volume taken from the staring concentration.

Line 157: Describe library normalization/pooling procedures. Was it done through quantitation or a normalization kit? Were DNA extraction blanks and negative controls also included in the final sequencing pool?

Lines 171: More frequently, we are reevaluating the casual use of read abundance filtering (i.e., minimum copy thresholds, relative read abundance thresholds). Even conservative thresholds < 1% can lead to surprising data loss and influence statistical trends. For a simple list of dietary detections (taxonomic analysis), I think these thresholds are ok. However, I would recommend reconsidering this 1% threshold for any analysis of diversity. Selection of these thresholds should be empirically supported by controls and datasets should be evaluated to determine if trends are sensitive to these thresholds. See the following publications for both simulation-based, empirical, and applied insight. Also keep in mind, based on the four publications below, that abundance-weighted metrics (e.g., Shannon’s diversity, Bray-Curtis) are robust to low frequency ASVs and may be used without using a read abundance threshold.

Littleford-Colquhoun BL, Freeman PT, Sackett VI, Tulloss CV, McGarvey LM, Geremia C, Kartzinel TR. 2022. The precautionary principle and dietary DNA metabarcoding: commonly used abundance thresholds change ecological interpretation. Molecular Ecology. 31(6):1615–1626. doi:10.1111/mec.16352.

Tercel MPTG, Cuff JP. The complex epistemological challenge of data curation in dietary metabarcoding: comment on “The precautionary principle and dietary DNA metabarcoding: commonly used abundance thresholds change ecological interpretation” by Littleford-Colquhoun et al. (2022). Molecular Ecology. n/a(n/a). doi:10.1111/mec.16576. [accessed 2022 Oct 24]. https://onlinelibrary.wiley.com/doi/abs/10.1111/mec.16576.

Littleford‐Colquhoun BL, Sackett VI, Tulloss CV, Kartzinel TR. 2022 Oct 20. Evidence‐based strategies to navigate complexity in dietary DNA metabarcoding: A reply. Molecular Ecology.:mec.16712. doi:10.1111/mec.16712.

Sanchez DE, Dikeman AL, Lyman JA, Zahratka J, Fofanov V, Walker FM, Chambers CL. 2024 Apr 20. Forbs, graminoids, and lepidopterans: breadth and seasonal variation in the diet of the New Mexico jumping mouse (Zapus luteus). Journal of Mammalogy.:gyae026. doi:10.1093/jmammal/gyae026.

Line 174: While generally required for diversity analysis, even-read depths are not required for generating a list of taxa because it can lead to unnecessary data loss. Why not use all reads for listing out dietary taxa? I recommend re-doing this analysis without even read depth. Otherwise, please justify its use.

I recommend omitting singleton ASVs from the ASV table, according to the reference below:

Ando H, Fujii C, Kawanabe M, Ao Y, Inoue T, Takenaka A. 2018. Evaluation of plant contamination in metabarcoding diet analysis of a herbivore. Scientific Reports. 8(1):15563. doi:10.1038/s41598-018-32845-w.

Lines 209-210: Please clarify the procedures and software used for processing the Sanger-derived sequences.

Lines 222-223: I also recommend using terminology from the software (e.g., percent identity). Although homology score is correct, perhaps it could be confused with bit score. Query cover is also an important parameter to consider unless all reference sequences spanned the entire fragment length of the amplicon. Otherwise, 98% ID will not be the same for reference sequences partially incorporating the amplicon region.

Line 231: Please reword for clarity. Instead of “accuracy”, perhaps use “the proportion of” and also rewrite to include the other taxonomic levels you provide this metric for (genus, family…).

Line 233: Even if plant-derived PCR inhibitors did not affect read abundance, would the read count still be meaningful? Contradictory to the authors’ assertion, the 1% abundance threshold used in pre-processing also assumes that the number of reads are meaningful. Fig 3 also depends on the read counts being meaningful. There are also a number of factors that influence read abundance from ingestion, digestion, to DNA extraction, to the PCR primers, to the bioinformatics pipeline. Visit the Clare et al. 2014 review above. I do not think read counts are inherently impractical because abundance-weighted diversity metrics, for example, down-weight low abundance features but not eliminating them. See Deagle et al. 2019 (below):

Deagle BE, Thomas AC, McInnes JC, Clarke LJ, Vesterinen EJ, Clare EL, Kartzinel TR, Eveson JP. 2019. Counting with DNA in metabarcoding studies: how should we convert sequence reads to dietary data? Molecular Ecology. 28(2):391–406. doi:10.1111/mec.14734.

Results:

Lines 253-255: This should be stated in either the methods or in the discussion.

Line 258: Remind the reader of the total sample size.

Lines 261-263: “Following rarefaction, we recovered 201 ASVs.”

Lines 270-276: This content seems like it should be in the “Identification of food plant resources” section. Be careful about interpreting detections and non-detections as diet items and non-diet items. For example, some plants detected in the feces may be accidental (they drank water that had pollen in it), incidental (they chewed on some bark but did not ingest the nutritious layer or constructed a daynest), or secondary ingestions (they ate an insect that consumed a plant). These ingestions can still be biologically relevant, however, but take care in interpreting metabarcoding data.

In terms of non-detection, no metabarcoding marker perfectly detects due to priming biases and other factors. That is why running multiple markers is important. Additionally, decisions made in the bioinformatics pipeline, such as the 1% abundance threshold and rarefaction, could have accidentally removed these species from the analysis. Thus, we cannot fully conclude non-consumption. The discussion section should include a biological interpretation on why these detections could have been used as dietary resources.

A more minor question is how do we know a portion of the bait (H. vulgare) was being avoided if these features were removed from the dataset during pre-processing?

Lines 281-282: These numbers could be low. Think about the 1% abundance threshold and the rarefaction.

Line 293: Use “individuals” or “individual mice” instead of fecal samples

Lines 295-298: Statistical support needed. Are these taxa associated with each month in a season?

Lines 297-298: These results are missing methods, as well as hypotheses/predictions/objectives in the introduction. Because the samples were collected among three different years, it could be that the different years influenced the trend. Perhaps treat year as a random effect or main effect if there are enough samples. Analysis of the plant composition will be a necessary. With respect to different months, I recommend an ordination, a group significance test (ANOSIM, permanova), and an ASV association test. R package vegan can do the first two. R package indicspecies can do the last analysis. However, be mindful about the specific tests you select and be mindful of their assumptions.

Lines 303-304: Please include methods for these results in the method section regarding the rarefaction analysis. Please elaborate on what coverage means, particularly in the results section.

Discussion:

Lines 338-229: This paragraph largely repeats information in the results. Instead, reintroduce the scope of the study. Tell us what these results mean according to the question, hypothesis, and predictions. Relate to the biology of this animal in relation to the artificial green space, not just other metabarcoding studies.

Lines 357-362: I think the biggest difference in the accuracy of rbcL is because trnL has terrible taxonomic resolution compared to rbcL. Similarly, this could also explain why you resolved more dietary taxa with rbcL than others did with trnL. If more fecal material means more taxa, perhaps there are other studies that have looked into that. This was not specifically tested in the study. Another consideration is that in a poor nutritious landscape, animals will generalize more and feed on poorer quality foods.

Line 375: A statistical test will be useful here. Be sure to reference the temporal scale (throughout all three years, each year). This is also a good place to reference why they could be consuming Cinnamomum, what part of the plant throughout a season. Is it because of competition with other individuals eating Quercus? Is there easier access to this food resource than Quercus?

Lines 381-382: A statistical test is needed as well as further elaboration.

Lines 389-390: Statistical support needed, specifically evaluating the main food plants, not just taxonomic richness.

Lines 395-402: The interpretation here is that green spaces should be invigorated with plants that were frequently identified in the diet. This seems rash and requires scrutiny. One of the goals was to identify plant resources to better design or maintain green spaces. But if these animals are consuming a variety of low quality foods, it might be counter-productive to plant more of these foods and instead focus on those of higher quality.

Further recommendations for discussion: consider weaknesses or alternative explanations for the findings. Consider the merits of a global reference library vs a local reference library for rbcL. Can dietary taxa be monitored in other urban green spaces in the area and will this local reference library still be useful? Are there any future recommendations for using this reference library or studying this animal’s diet in green spaces or in natural settings?

Conclusions:

Lines 405-413: I'm not sure this data describes the feeding habits (e.g., foraging behaviors), rather, it recovered dietary taxa. The local reference library gave better species level resolution than previous methods. Are there other lines of information that could be obtained for more robust decision making? Are there any other green space studies or similar settings that can be referenced?

Figures and tables:

Table and Figure captions: Overall, we need more information and should be able to stand alone from the manuscript (e.g., reference to focal species and setting).

Fig 3: This visualization is not appropriate. There are too many categories and therefore too many colors for the human eye to differentiate. I also do not think it is valid to only include taxa only in the top 20% of the reads because there is likely important information in the lower end of the read counts. This also contradicts an earlier statement that presence/absence would only be considered due to amplification biases. Perhaps an ordination would be more appropriate to summarize seasonal changes in diet.

Fig 4: No need for multiple colors with the boxplots. Use more precise terminology such as taxonomic richness for y-axis.

6. PLOS authors have the option to publish the peer review history of their article (what does this mean? ). If published, this will include your full peer review and any attached files.

**Do you want your identity to be public for this peer review?** For information about this choice, including consent withdrawal, please see our Privacy Policy .

Reviewer #1: No

Reviewer #2: No

---

## [Author Response · Author response to Decision Letter 1]

31 Oct 2024

Dear Editor,

Thank you for your useful comments. The manuscript has been revised according to the reviewers' comments. The major change is the addition of a discussion on the seasonal differences in the diet in the Apodemus speciosus plant food resources, based on the results of NMDS and PerMANOVA analyses. Additionally, the foraging characteristics of the A. speciosus were compared with previous studies to provide further insights into the specific features of the study site. Responses to the reviewers' comments are given below.

Dear Reviewer #1

Thank you for your useful comments. Responses to your comments are given below.

Reviewer comment R1-1:

L46-60: The most important paragraph in introduction showing the motivation of this study is the 1st paragraph. However, almost all of references (9/12) are from papers written in Japanese. In this situation, it is difficult to convince worldwide researchers to understand that the artificial green space that can be regarded as a space used in OECM is important place. Cite more worldwide papers. If this idea began in Japan, please explain so and show that almost nothing has been conducted in the world except for Japan.

Response:

Revised manuscript: L44-68

We have corrected this sentence according to the Editors’ comment.

Reviewer comment R1-2:

Didn’t you examine negative control analyses, for example, at the DNA extraction or PCR steps? Such negative control PCR samples should be sequenced and treated appropriately in the sequence read matrix.

Response:

Revised manuscript: L501-514

We have added the paragraph to Discussion.

In this study, we did not implement negative and positive controls. We have added a discussion of this issues in the discussion section.

Reviewer comment R1-3:

This study is important because it provides information of diets of A. speciosus in an artificial environment in urban place and central part of Japan. However, there are no discussions in these points. How different are the diets of field mice in northern (Hokkaido), central (Aichi), and southern (Seto Inland Sea) parts of Japan? How different are the diets of field mice between urban and natural environments? These discussions should be made.

Response:

Revised manuscript: L353-381, 449-494

We have added comparison of identification of food plants results reported in previous reports. In addition, we have added the analysis of seasonal variations in study area.

Reviewer comment R1-4:

L64: inhabited > inhabits [this species is also inhabiting forest now]

Response:

Revised manuscript: L77

We have corrected this sentence according to the Editors’ comment.

Reviewer comment R1-5:

L69: I am not sure if only conserving preferred resources are good or not in terms of ecosystem conservation. It may be good for A. speciosus, but it must be considered that the ecosystem consists of diverse organisms. The idea shown here is too simplistic.

Response:

Revised manuscript: L83-86

We have corrected this sentence according to the Editors’ comment.

Reviewer comment R1-6:

L72-73: The sentence may not be accurate. There are several traditional studies using the morphological analyses of the stomach contents, which were conducted on the forest and around the cultivated field (considered artificial green space in the Satoyama [OECM] landscape). These studies also examined food plant resources of A. speciosus. So, you should cite these traditional papers if you would like to express so like the current sentence. Papers that you cited are a DNA barcoding study using subcloning method [17] and DNA metabarcoding studies [18-19]. In this case, you should add Sato et al. (2022: Mammal Research 67: 109–122) here.

Response:

Revised manuscript: L87-101

We have corrected these sentences according to the Editors’ comment.

We have added papers investigating food resources other than DNA metabarcoding of Apodemus speciosus. And we have also added the problem of classification resolution by DNA metabarcoding in analyzing food resources of A. speciosus.

Reviewer comment R1-7:

L74-77: All references are not from studies of A. speciosus. Because you started this paragraph from the story of A. speciosus, you should introduce such traditional studies of A. speciosus.

Response:

Revised manuscript: L87-120

We have corrected these sentence and references according to the Editors’ comment.

Reviewer comment R1-8:

L79: [20] is not a study of A. speciosus.

Response:

Revised manuscript: L92

We have corrected these references according to the Editors’ comment.

Reviewer comment R1-9:

L81: You should add Sato et al. (2022: Mammal Research 67: 109–122) here. In addition, [27-38] is not appropriately selected but seems to be selected randomly. You should cite only relevant papers, for example, papers on DNA metabarcoding method applied to rodents.

Response:

Revised manuscript: L92

We have corrected these references according to the Editors’ comment.

Reviewer comment R1-10:

L83: You should add Sato et al. (2022: Mammal Research 67: 109–122) here.

Response:

Revised manuscript: L94, 100, 425, 427, 435, 439, 453, 454, 464, 472, 475, 487, 494

We have added the references according to the Editors’ comment.

Reviewer comment R1-11:

L83-86: You may misunderstand between low resolution and low accuracy. Both studies showed accurate identifications but trnL mostly showed family level identification (low resolution) without a local plant list or DNA database. Change the expression.

Response:

Revised manuscript: L96, 97, 105, 106, 109, 112, 119, 245, 419, 427, 431, 432, 541

We have corrected these sentences according to the Editors’ comment.

Reviewer comment R1-12:

L87: Also consider the usage of accuracy.

Response:

Revised manuscript: L112

We have corrected this sentence according to the Editors’ comment.

Reviewer comment R1-13:

L89: [19] and Sato et al. (2022) above also used this marker.

Response:

This sentence has been deleted as part of revisions in the revised manuscript.

Reviewer comment R1-14:

L91: Just long is not enough for the reason to use this marker. Previously 7 of 27 papers of rodent dietary DNA metabarcoding studies used ITS2 marker (e.g., Murano et al. 2023 and Sato et al. 2023). Only two studies used rbcL marker (Latinne et al. 2014 Journal of Cave and Karst Studies 76: 139–145, Gabrielson et al. 2023 Biological Invasions, https://doi.org/10.1007/s10530-023-03159-4). ITS2 is longer than rbcL and probably more informative, but it may show less taxonomic coverage than rbcL. So, you must not oversimplify the background information. You should select markers in terms of length, informativeness (genetic variation), taxonomic coverage, or cost (long marker requires expensive sequencing kit), not just based on length.

Response:

Revised manuscript: L92-112

We have corrected this sentence according to the Editors’ comment.

Reviewer comment R1-15:

L131-132: What is the procedure “dried overnight at 60 degree” for? It seems to lead to DNA degradation. What do you think?

Response:

Revised manuscript: L160-163

We have corrected this sentence according to the Editors’ comment.

The reason for using dried feces was added.

Since we have not conducted the subject tests, we do not know the detailed damage to DNA caused by this operation. However, the results of electrophoresis of the total DNA showed that high molecular weight DNA was obtained.

Therefore, we do not think that there is any significant damage.

Fig. Total DNA electrophoresis results using the 60℃ drying method

This is the result of 0.8% agarose gel electrophoresis of a DNA sample extracted from feces dried at 60℃.

Although there is some smearing, high molecular weight DNA, presumed to be chloroplast DNA, is clearly visible.

Reviewer comment R1-16:

L141, 151: Write a full name of the kit KAPA HiFi.

Response:

Revised manuscript: L173, 175

We have corrected this sentence according to the Editors’ comment.

Reviewer comment R1-17:

L147-148: Did you perform “ligation” to put both indices and adaptors or put them through 2nd PCR? I suppose that the latter is true, but your explanation looks the opposite.

Response:

Revised manuscript: L181-185

We have corrected this sentence according to the Editors’ comment.

Reviewer comment R1-18:

L195: Consider the usage of accurate.

Response:

Please see the reply to R1-11 for details.

We have corrected this sentence according to the Editors’ comment.

Reviewer comment R1-19:

L224, 231: Consider the usage of accuracy.

Response:

Please see the reply to R1-11 for details.

We have corrected this sentence according to the Editors’ comment.

Reviewer comment R1-20:

L251-255: Why were these plants excluded. At least genus level discussion can be made. Even if these species grow near the capture site, contamination could not necessarily occur. What is the mechanism for the contamination. Pollen? The reason for the exclusion is not clear.

Response:

Revised manuscript: L322-325, L527-533

We have corrected this sentence according to the Editors’ comment.

The reason for its exclusion is that it cannot be distinguished from the DNA sequence derived from "captured bait."

Reviewer comment R1-21:

L336, 354, 360: Consider the usage of accuracy.

Response:

Please see the reply to R1-11 for details.

We have corrected this sentence according to the Editors’ comment.

Reviewer comment R1-22:

L360-362: You used rbcL for its longer sequence (as noted in Introduction), but there are no discussions regarding this length (variation) issue. Also discuss informativeness of these markers. Because ITS2 has been the most frequently used markers for plant dietary species, you should also discuss the usefulness of rbcL compared with the ITS2 (UniPlant marker).

Response:

Revised manuscript: L92-120, 428-433

We have corrected this sentence according to the Editors’ comment.

We added the reason why we used rbcL in this study.

Reviewer comment R1-23:

L369-372: See Sato et al. (2022) in Mammal Research above and discuss with it regarding Lauraceae.

Response:

Revised manuscript: L451-494

We have corrected this sentence according to the Editors’ comment.

Reviewer comment R1-24:

L373: You should add Sato et al. (2022: Mammal Research 67: 109–122) here.

Response:

Revised manuscript: L485-487

We have corrected this sentence according to the Editors’ comment.

Reviewer comment R1-25:

L379: You assumed “seeds”, but looking at Fig. 3, Fagaceae or Quercus is not so abundant in autumn. Is it correct to say that these are from acorns?

Response:

Revised manuscript: L462-494

We have corrected this sentence according to the Editors’ comment.

We added description foraging of Cinnamomum spp.

Reviewer comment R1-26:

L389-402: Sato et al. (2018) and Sato et al. (2022) showed seasonal variations of the diet of A. speciosus in Hokkaido and Seto Inland Sea, respectively. Similar trend in the diet of this species among seasons is not usual for this species. You should discuss this point and assess if this unusual trend was caused by an artificial environment or not.

Response:

Revised manuscript: L292-310, L365-371, L453-461, Fig3

The NMDS analysis and PerMANOVA analysis were added according to the opinions in R2-27, R2-31, R2-32, R2-38.

In addition, We have added sentences to the discussion.

Reviewer comment R1-27:

L401-402: Why did you think Quercus is needed in the study place? This species could live in the place for more than 50 years. This suggestion has no rationale.

Response:

This sentence has been deleted as part of revisions in the revised manuscript.

Reviewer comment R1-28:

L464: Iwase > Iwasa

Response:

Revised manuscript: L625

We corrected to "Iwasa" according to the Editors’ comment.

Reviewer comment R1-29:

L480: Apodemus should be italic.

Response:

Revised manuscript: L671

We corrected "Apodemus" to italic, according to the Editors’ comment.

Dear Reviewer #2

Thank you for your useful comments. Responses to your comments are given below.

Reviewer comment R2-1:

Line 73: Be careful with the word “preference”. We can determine what an animal uses or selects within particular spatial and temporal scales but can we really know what an animal prefers? I recommend the reference below.

Thomas DL, Taylor EJ. 1990. Study Designs and Tests for Comparing Resource Use and Availability. The Journal of Wildlife Management. 54(2):322–330. doi:10.2307/3809050.

Response:

Revised manuscript: L98-101

We corrected from "food plant preference" to "food plant resources".

Reviewer comment R2-2:

Lines 68-86: This content must better describe what we know about the animal’s phenology and diet, for example, in undisturbed or less disturbed habitat. Do they consume foliage, seeds, roots … etc.? Are they strict herbivores? Are they seed predators, do they disperse seeds? Most rodents are omnivores in order Rodentia. If A. speciosus is an omnivore, I suggest better justifying why you only investigate herbivory and not other trophic levels. If the animal consumes insects, can they assist with pest control, for example? Could any of the plants detected in the feces actually be plants that an insect consumed? Do they hibernate? Do they cache seeds for the winter? Overall, this content should be much improved by delving deeper into the study system and what the species would need if placed in an environment of human design.

Response:

Revised manuscript: L69-120

We have corrected these sentences according to the Editors’ comment.

We have added an explanation in the main text about the Apodemus speciosus being an omnivore and the significance of analyzing food plant resources.

As mentioned in the main text, the reason for investigating food plant resources is to explore their potential application in artificial green space management. Since the population of the Apodemus speciosus fluctuates based on the availability of food plant resources, such as acorns, vegetation management can have a significant impact on their population dynamics.

Reviewer comment R2-3:

Lines 87-93: It seems like the authors present a false dichotomy between trnL and rbcL. Multiple markers are better than one and barcodes other than trnL and rbcL are available for plant metabarcoding, such as ITS2. ITS2 is also commonly used and delivers even better taxonomic resolution than rbcL. My main recommendation would be to also include ITS2 in the dataset, if funding permits; however, I would at least mention ITS2 and make a better case for only using rbcL. I hope the authors find the two publications below helpful.

Fahner NA, Shokralla S, Baird DJ, Hajibabaei M. 2016. Large-Scale Monitoring of Plants through Environmental DNA Metabarcoding of Soil: Recovery, Resolution, and Annotation of Four DNA Markers. PLOS ONE. 11(6):e0157505. do

---

## [Decision Letter · Decision Letter 1]

19 Nov 2024

PONE-D-24-12099R1Identification of Large Japanese field mouse *Apodemus speciosus* food plant resources in an industrial green space using DNA metabarcodingPLOS ONE

Dear Dr. Fujii,

Thank you for submitting your manuscript to PLOS ONE. After careful consideration, we feel that it has merit but does not fully meet PLOS ONE’s publication criteria as it currently stands. Therefore, we invite you to submit a revised version of the manuscript that addresses the points raised during the review process.

**ACADEMIC EDITOR** :

My general recommendation is Revision with minor issues. I need to see further replies from the authors to evaluate this manuscript more correctly.

. 

We look forward to receiving your revised manuscript.

Kind regards,

Faham Khamesipour, Ph.D.

Academic Editor

PLOS ONE

Journal Requirements:

Reviewers' comments:

Reviewer's Responses to Questions

**Comments to the Author**

1. If the authors have adequately addressed your comments raised in a previous round of review and you feel that this manuscript is now acceptable for publication, you may indicate that here to bypass the “Comments to the Author” section, enter your conflict of interest statement in the “Confidential to Editor” section, and submit your "Accept" recommendation.

Reviewer #1: (No Response)

Reviewer #2: (No Response)

2. Is the manuscript technically sound, and do the data support the conclusions?

Reviewer #1: Yes

Reviewer #2: Partly

3. Has the statistical analysis been performed appropriately and rigorously?

Reviewer #1: Yes

Reviewer #2: No

4. Have the authors made all data underlying the findings in their manuscript fully available?

Reviewer #1: Yes

Reviewer #2: Yes

5. Is the manuscript presented in an intelligible fashion and written in standard English?

Reviewer #1: No

Reviewer #2: Yes

6. Review Comments to the Author

Reviewer #1: The manuscript has been improved by your correction based on the reviewer’s comments. However, I still have concerns on this manuscript as described below. I would like to have your replies again.

Major concerns

1. I still cannot understand what the factor affecting the difference between this study and previous studies regarding the seasonal dietary changes is. Your should discuss more about the reason why you did not detect remarkable seasonal changes in diets compared with the previous studies.

Specific concerns

P2L29: feces sample > fecal sample

P2L37-38: It would be more important to show that your vegetation survey and your sequencing the rbcL gene for 104 dominant plant species were performed to construct a local plant database including 651 plant species of the 796 plants growing in the study site.

P4L74 and P4L80: [23] would not be an appropriate reference which is based on the tundra ecosystem and not related to A. speciosus.

P4L84: contribute > provide

P5L97-97: food resources > food plant resources

P5L110-111: I still have concerns on the reason why you did not use ITS regions for this study. You explained that the ITS has not been widely applied. However, there have been 27 dietary DNA metabarcoding studies for rodents using the plant markers to date. Among them, 12 papers used ITS2, 19 papers used trnL, and only 2 papers used rbcL marker. Especially in Japan, Murano et al. (2023; https://doi.org/10.3106/ms2023-0015), Sato et al. (2023; https://doi.org/10.3106/ms2023-0003), and Kobayashi et al. (2024; https://doi.org/10.3106/ms2024-0007) used the ITS marker for several rodent species. What you explained here is not consistent with the rodent DNA metabarcoding studies. You should describe the current situation more correctly, citing these relevant papers.

P6L125: Fig. 1 > Add longitude and latitude in the left figure.

P7L149: feces sample > fecal sample

P7L150-151: How long it takes between your sampling in the field and preservation in the freezer in your laboratory and how (under what temperature) you kept your sample in 1.5 ml tube during the transportation should be described.

P8L168-169: Probably in the iSeq platform, the adjacent sequencing position would be separated, so that this problem is not likely to occur. Not all the Illumina sequencing platform are caused by this problem. Please check it and be careful of the expression.

P11L222: What number was the minimum read number among samples to determine the number 1000 for analyses? Please note it if possible.

P13L254: feces sample > fecal sample

P14L268: feces sample > fecal sample

P15L296: > relative read abundance

P17L325: > to bait or plants growing at the study site.

P18L350: Is “98% percent identity” a correct expression? Probably, “percent” would not be necessary.

P19L358: across of? Write correct English.

P19L362: Note here that Quercus is a member of Fagaceae.

P19L367-369: How about the result if you separate winter and the other seasons? Fig. 4 and 5 showed the difference between them.

P19L372: generated compare? Write correct English.

P28L423: [37] was a report from June to September in Hokkaido, [38] was a report from February to May in the Seto Inland Sea, and [39] was a report from March to August in the Seto Inland Sea. Information here should be wrong.

P28L435-P29437: “the number of fecal samples is small” cannot be a reason for excluding this study for comparison. Instead, small numbers would be related to fewer detection of plant taxa. This is a point that should be discussed.

P2pL439: 60 mg is so specific value. Please show the data to support this statement.

P29L446: feces sample > fecal sample

P30L459-461: I cannot understand why vegetation environment was not favorable for A. speciosus despite of the presence of more than 700 plant species. What may be the reason?

P31L486: Discussion > discussion

P32L499: What is “seasonal forage plants”? Are they plants that A. speciosus could eat in all the season? Or combination of various kinds of plants that were consumed by A. speciosus in different seasons?

Reviewer #2: Having reviewed the original submission of this manuscript, I am pleased with the efforts the authors made in the current revision. I am particularly satisfied with the statistical analyses and the details the authors included about the life history of their focal species. Nevertheless, there are several points that were not sufficiently addressed.

I am most concerned with the authors’ responses clarifying that negative controls were omitted from the experimental design. In case there was a translation issue in the authors’ response, if the authors did indeed include negative controls in their library preparation but did not sequence them, I will be ok with that, assuming there is no strong evidence of amplification in other data sources. That said, the authors must provide evidence for not including the negative controls in the sequencing pool, such as qPCR data from the normalization procedures. But in the future, always ensure negative controls (at as many levels as possible) are sequenced along with the samples.

If no negative controls were included in library preparation, I highly recommend the authors redo the sequencing dataset starting from PCR and include multiple PCR negative controls. I understand that this is a harsh recommendation. I also understand a lot of hard work went into preparing the existing dataset. But I find omission of this basic experimental component too concerning. While the authors provide a section recognizing potential laboratory contaminants, this section does not and probably cannot address whether cross-contamination occurred. Such contaminants can occur at high read counts, above the 1% read threshold, particularly if cross-contamination was egregious. Without negative controls, there is not a great gauge for this and so any analysis that compares samples or groups of samples is difficult to interpret biologically.

Minor comments:

General: When referring to ITS, refer specifically to ITS2 throughout.

Lines 108-109: Regarding increased cost of multiple markers, the authors should address that multiplex PCR protocols exist for co-amplification of rbcL and ITS2. This allows both markers to be processed for the same price as one. Here is a paper that describes this:

Sanchez DE, Dikeman AL, Lyman JA, Zahratka J, Fofanov V, Walker FM, Chambers CL. 2024. Forbs, graminoids, and lepidopterans: breadth and seasonal variation in the diet of the New Mexico jumping mouse (Zapus luteus). Journal of Mammalogy. 105(4):880–898. doi:10.1093/jmammal/gyae026.

Lines 212-213: Non-dietary taxa can still occur at levels above this threshold, as Ando et al. (2018) also discuss. Newer science shows that these thresholds can strongly affect diversity statistics and so I recommend the authors repeat their statistical analysis of ASVs without a read threshold to determine if the threshold could be artificially influencing the results. In response to the authors’ rebuttal, I am not suggesting that the threshold be taken out but that it be validated to ensure the threshold is not affecting statistical results. This additional analysis can be supplementary material and will add sufficient statistical rigor to this work. I provide reference for further information regarding the effects on diversity statistics.

Littleford-Colquhoun BL, Freeman PT, Sackett VI, Tulloss CV, McGarvey LM, Geremia C, Kartzinel TR. 2022. The precautionary principle and dietary DNA metabarcoding: commonly used abundance thresholds change ecological interpretation. Molecular Ecology. 31(6):1615–1626. doi:10.1111/mec.16352.

Lines 283-285: Please use precise terminology: “frequency of occurrence” or “proportion/percent frequency of occurrence”. See the citation below:

Deagle BE, Thomas AC, McInnes JC, Clarke LJ, Vesterinen EJ, Clare EL, Kartzinel TR, Eveson JP. 2019. Counting with DNA in metabarcoding studies: how should we convert sequence reads to dietary data? Molecular Ecology. 28(2):391–406. doi:10.1111/mec.14734.

Lines 285-287: This sentence is still difficult to understand. I recommend saying “We summarized taxonomic resolution of each ASV/taxon by …”, rather than describing “accuracy”.

Line 309: Is this meant to be alpha = 0.05? Otherwise, the results of this statistical analysis should be in the results.

Line 368: Report the actual P-values.

Line 505: Identifying taxa occurring in negative controls does not necessarily mean that those sequences need to be eliminated from the dataset. If a high concentration or high frequency diet item contaminates another sample, it might not be wise to omit it.

7. PLOS authors have the option to publish the peer review history of their article (what does this mean? ). If published, this will include your full peer review and any attached files.

**Do you want your identity to be public for this peer review?** For information about this choice, including consent withdrawal, please see our Privacy Policy .

Reviewer #1: No

Reviewer #2: No

---

## [Author Response · Author response to Decision Letter 2]

20 Dec 2024

Dear Editor,

Thank you for your useful comments. The manuscript has been revised according to the reviewers' comments.

Dear Reviewer #1

Thank you for your useful comments. Responses to your comments are given below.

Reviewer comment R1-1:

I still cannot understand what the factor affecting the difference between this study and previous studies regarding the seasonal dietary changes is. Your should discuss more about the reason why you did not detect remarkable seasonal changes in diets compared with the previous studies.

Response:

Revised manuscript: L

We have added a discussion and Support_Figure1 regarding the vegetation information surrounding each study site.

Further discussion was considered inconclusive without a detailed comparison of vegetation and carries the risk of over-discussion.

Reviewer comment R1-2:

P2L29: feces sample > fecal sample

Response:

Revised manuscript: L29

We have corrected this sentence according to the Editors’ comment.________________________________________

Reviewer comment R1-3:

P2L37-38: It would be more important to show that your vegetation survey and your sequencing the rbcL gene for 104 dominant plant species were performed to construct a local plant database including 651 plant species of the 796 plants growing in the study site.

Response:

Revised manuscript: L36-39

We have corrected this sentence according to the Editors’ comment.

Reviewer comment R1-4:

P4L74 and P4L80: [23] would not be an appropriate reference which is based on the tundra ecosystem and not related to A. speciosus.

Response:

Revised manuscript: L73-74, 81

We have corrected the reference and made corresponding adjustments to the main text according to the Editors’ comment.

Reviewer comment R1-5:

P4L84: contribute > provide

Response:

Revised manuscript: L85

We have corrected this sentence according to the Editors’ comment.

Reviewer comment R1-6:

P5L97-97: food resources > food plant resources

Response:

Revised manuscript: L98-99

We have corrected this sentence according to the Editors’ comment.

Reviewer comment R1-7:

P5L110-111: I still have concerns on the reason why you did not use ITS regions for this study. You explained that the ITS has not been widely applied. However, there have been 27 dietary DNA metabarcoding studies for rodents using the plant markers to date. Among them, 12 papers used ITS2, 19 papers used trnL, and only 2 papers used rbcL marker. Especially in Japan, Murano et al. (2023; https://doi.org/10.3106/ms2023-0015), Sato et al. (2023; https://doi.org/10.3106/ms2023-0003), and Kobayashi et al. (2024; https://doi.org/10.3106/ms2024-0007) used the ITS marker for several rodent species. What you explained here is not consistent with the rodent DNA metabarcoding studies. You should describe the current situation more correctly, citing these relevant papers.

Response:

Revised manuscript: L110-126

We think no reason to limit the DNA region of analysis of DNA barcoding to rodent food analysis methods.

As reason for DNA barcoding region selection, we focused on the 796 species found in the study area and searched the number of registrations to NCBI in the rbcL, ITS2 and trnL regions using NCBI web site. Taxonomy ID and analysis DNA region name were used for the search.

Search method:

Search the following keywords at NCBI

txidXXXX[Organism:exp] AND YYYY[All Fields]

XXXX: taxonomi ID

YYYY: DNA region name (rbcL, transcribed+spacer+2, tRNA-Leu)

The results of this analysis showed that the number of registrations for rbcL was the highest. This result was added to the method section to reinforce the justification for using rbcL region.

Reviewer comment R1-8:

P6L125: Fig. 1 > Add longitude and latitude in the left figure.

Response:

Revised manuscript: Fig. 1

We have corrected these references according to the Editors’ comment.

Reviewer comment R1-9:

P7L149: feces sample > fecal sample

Response:

Revised manuscript: L159

We have corrected these references according to the Editors’ comment.

Reviewer comment R1-10:

P7L150-151: How long it takes between your sampling in the field and preservation in the freezer in your laboratory and how (under what temperature) you kept your sample in 1.5 ml tube during the transportation should be described.

Response:

Revised manuscript: L160-162

Samples were stored at room temperature (3-5 hours) from the time of collection until stored in the laboratory freezer.

We have corrected this sentence according to the Editors’ comment.

Reviewer comment R1-11:

P8L168-169: Probably in the iSeq platform, the adjacent sequencing position would be separated, so that this problem is not likely to occur. Not all the Illumina sequencing platform are caused by this problem. Please check it and be careful of the expression.

Response:

We have received a response from Illumina's official support stating that, even for patterned flow cells on the iSeq, considerations regarding base diversity should be taken into account.

Please also refer to the following URL.

https://knowledge.illumina.com/instrumentation/general/instrumentation-general-reference_material-list/000001527

Improving nucleotide diversity is necessary even for iSeq.

Reviewer comment R1-12:

P11L222: What number was the minimum read number among samples to determine the number 1000 for analyses? Please note it if possible.

Response:

Revised manuscript: L233

We added Column (Low frequency removal reads) to Support_Table4 and Support_Figure2 in this sentence according to the Editors’ comment.

Reviewer comment R1-13:

P13L254: feces sample > fecal sample

Response:

Revised manuscript: L265

We have corrected this sentence according to the Editors’ comment.

Reviewer comment R1-14:

P14L268: feces sample > fecal sample

Response:

Revised manuscript: L279

We have corrected this sentence according to the Editors’ comment.

Reviewer comment R1-15:

P15L296: > > relative read abundance

Response:

Revised manuscript: L307

We have corrected this sentence according to the Editors’ comment.

Reviewer comment R1-16:

P17L325: > to bait or plants growing at the study site.

Response:

Revised manuscript: L336

We have corrected this sentence according to the Editors’ comment.

Reviewer comment R1-17:

P18L350: Is “98% percent identity” a correct expression? Probably, “percent” would not be necessary.

Response:

Revised manuscript: L362

We have corrected this sentence according to the Editors’ comment.

Reviewer comment R1-18:

P19L358: across of? Write correct English.

Response:

Revised manuscript: L3370

We have corrected this sentence according to the Editors’ comment.

Reviewer comment R1-19:

P19L362: Note here that Quercus is a member of Fagaceae.

Response:

Revised manuscript: L3372

The details of Fagaceae-1 and Quercus are provided in Table 1.

The main text has been corrected to clarify that these are identified taxon names in this study.

Reviewer comment R1-20:

P19L367-369: How about the result if you separate winter and the other seasons? Fig. 4 and 5 showed the difference between them.

Response:

The rationale for comparing winter and other seasons is unclear, and there is also a bias due to the uneven number of fecal samples collected. Therefore, this additional analysis was not performed.

Reviewer comment R1-21:

P19L372: generated compare? Write correct English.

Response:

Revised manuscript: L385

We have corrected this sentence according to the Editors’ comment.

Reviewer comment R1-22:

P28L423: [37] was a report from June to September in Hokkaido, [38] was a report from February to May in the Seto Inland Sea, and [39] was a report from March to August in the Seto Inland Sea. Information here should be wrong.

Response:

Revised manuscript: L434-442

We have corrected this sentence according to the Editors’ comment.

Reviewer comment R1-23:

P28L435-P29437: “the number of fecal samples is small” cannot be a reason for excluding this study for comparison. Instead, small numbers would be related to fewer detection of plant taxa. This is a point that should be discussed.

Response:

This comparison is based on the number of detections plant taxa per fecal sample, so there is no correlation between the number of analyzed fecal samples and the detection count.

However, "Sato et al. [37]" refers to a cumulative sample size of 12, which is limited compared to other studies. Considering the variability in the number of detected plant species per fecal sample, it was deemed better to exclude it from the analysis.

Reviewer comment R1-24:

P2pL439: 60 mg is so specific value. Please show the data to support this statement.

Response:

There is no clear basis for the 60 mg. We determined that 60 mg is appropriate based on the fecal weight collected in this study.

Reviewer comment R1-25:

P29L446: feces sample > fecal sample

Response:

Revised manuscript: L461

We have corrected this sentence according to the Editors’ comment.

Reviewer comment R1-26:

P30L459-461: I cannot understand why vegetation environment was not favorable for A. speciosus despite of the presence of more than 700 plant species. What may be the reason?

Response:

Revised manuscript: L485-490

In this study area, about 700 species have been identified, however the primary plants are the 104 species we collected. To clarify the causes of the foraging characteristics of winter in this study, detailed vegetation survey results, including those from previously reported study sites [37, 38, 39], are necessary. Therefore, with the current information, further discussion would result in overinterpretation, so we have limited our description to the facts derived from the results.

Reviewer comment R1-27:

P31L486: Discussion > discussion

Response:

Revised manuscript: L506

We have corrected this sentence according to the Editors’ comment.

Reviewer comment R1-28:

P32L499: What is “seasonal forage plants”? Are they plants that A. speciosus could eat in all the season? Or combination of various kinds of plants that were consumed by A. speciosus in different seasons?

Response:

Revised manuscript: L519

We have corrected this sentence according to the Editors’ comment.

Dear Reviewer #2

Thank you for your useful comments. Responses to your comments are given below.

Reviewer comment R2-1:

General: When referring to ITS, refer specifically to ITS2 throughout.

Response:

Revised manuscript: L104, 107, 120, 121, 124

We corrected from "food plant preference" to "food plant resources".

Reviewer comment R2-2:

Lines 108-109: Regarding increased cost of multiple markers, the authors should address that multiplex PCR protocols exist for co-amplification of rbcL and ITS2. This allows both markers to be processed for the same price as one. Here is a paper that describes this:

Sanchez DE, Dikeman AL, Lyman JA, Zahratka J, Fofanov V, Walker FM, Chambers CL. 2024. Forbs, graminoids, and lepidopterans: breadth and seasonal variation in the diet of the New Mexico jumping mouse (Zapus luteus). Journal of Mammalogy. 105(4):880–898. doi:10.1093/jmammal/gyae026.

Response:

When conducting DNA metabarcoding, we outsource the task to IDEA Corporation. Indeed, it is possible to analyze multiple regions in a single run using next-generation sequencers. However, since our laboratory outsources the work, costs are incurred depending on the number of samples.

We hope for your understanding.

Reviewer comment R2-3:

Lines 212-213: Non-dietary taxa can still occur at levels above this threshold, as Ando et al. (2018) also discuss. Newer science shows that these thresholds can strongly affect diversity statistics and so I recommend the authors repeat their statistical analysis of ASVs without a read threshold to determine if the threshold could be artificially influencing the results. In response to the authors’ rebuttal, I am not suggesting that the threshold be taken out but that it be validated to ensure the threshold is not affecting statistical results. This additional analysis can be supplementary material and will add sufficient statistical rigor to this work. I provide reference for further information regarding the effects on diversity statistics.

Response:

Revised manuscript: L381

And, we have added the NMDS analysis results utilizing all ASV information as a supporting figure2.

We understand that changes in the threshold affect alpha and beta diversity. However, we recognize that this issue is still under discussion Tercel et al. 2022, doi: 10.1111/mec.16576�.

In this study, we did not include negative controls. Therefore, as noted in the text, contamination is present. No significant differences were observed in the beta diversity analysis (NDMS and PerMANOVA) with or without ASV filtering (Fig. 3 and Supporting Figure 2). When compared threshold-based filtering and no filtering, we consider threshold-based filtering to be appropriate for this study.

Reviewer comment R2-4:

Lines 283-285: Please use precise terminology: “frequency of occurrence” or “proportion/percent frequency of occurrence”. See the citation below:

Response:

Revised manuscript: L294

We have corrected this sentence according to the Editors’ comment.

Reviewer comment R2-5:

Lines 285-287: This sentence is still difficult to understand. I recommend saying “We summarized taxonomic resolution of each ASV/taxon by …”, rather than describing “accuracy”.

Response:

Revised manuscript:L〇-〇

We have corrected this sentence according to the Editors’ comment.

Reviewer comment R2-6:

Line 309: Is this meant to be alpha = 0.05? Otherwise, the results of this statistical analysis should be in the results.

Response:

Revised manuscript: L312, 320-321

We have corrected this sentence according to the Editors’ comment.

Reviewer comment R2-7:

Line 368: Report the actual P-values.

Response:

Revised manuscript: L379-380

We have corrected this sentence according to the Editors’ comment.

Reviewer comment R2-8:

Line 505: Identifying taxa occurring in negative controls does not necessarily mean that those sequences need to be eliminated from the dataset. If a high concentration or high frequency diet item contaminates another sample, it might not be wise to omit it.

Response:

Revised manuscript: L525

We have corrected this sentence according to the Editors’ comment.

---

## [Decision Letter · Decision Letter 2]

13 Jan 2025

PONE-D-24-12099R2Identification of Large Japanese field mouse *Apodemus speciosus* food plant resources in an industrial green space using DNA metabarcodingPLOS ONE

Dear Dr. Fujii,

Thank you for submitting your manuscript to PLOS ONE. After careful consideration, we feel that it has merit but does not fully meet PLOS ONE’s publication criteria as it currently stands. Therefore, we invite you to submit a revised version of the manuscript that addresses the points raised during the review process.

**ACADEMIC EDITOR:**

It is almost acceptable with minor corrections.

The experimental design is sound, though it could be further strengthened by covering two or three consecutive years or ensuring equal sampling efforts across all time frames.

The use of the rbcL marker is well-justified, and the sequencing, trimming, and data analysis are thoroughly described. Additionally, the authors have effectively addressed and responded to the reviewers’ comments, which has significantly enhanced the clarity and quality of the manuscript.

However, I have a few minor corrections to suggest to further improve the manuscript’s clarity for readers. Furthermore, while the authors revised lines 37–38 based on Reviewer 1's comments, I disagree with the change. I believe the original sentence provided by the authors better emphasized the importance of creating a local database. Including the number of sequences in the database detracts from the broader significance of this statement.

Line 134: consists of -> is located in

Line 135 - 140: Required English edition for clarity

Line: 148: suffering -> animal suffering

Line 150: and -> with

Line 159 -> minimize

Line 254: DNA databank -> database

Line 254-355: and that this -> which

Line 298: by -> relative to

Line 374: after -> following

Line 374: are identifiers used to -> serve as identifiers to

Line 392 - 393: the statement is correct however you should mention that the number of winter samples was limited

. ==============================

We look forward to receiving your revised manuscript.

Kind regards,

Faham Khamesipour, Ph.D.

Academic Editor

PLOS ONE

Journal Requirements:

Additional Editor Comments:

It is almost acceptable with minor corrections.

The experimental design is sound, though it could be further strengthened by covering two or three consecutive years or ensuring equal sampling efforts across all time frames.

The use of the rbcL marker is well-justified, and the sequencing, trimming, and data analysis are thoroughly described. Additionally, the authors have effectively addressed and responded to the reviewers’ comments, which has significantly enhanced the clarity and quality of the manuscript.

However, I have a few minor corrections to suggest to further improve the manuscript’s clarity for readers. Furthermore, while the authors revised lines 37–38 based on Reviewer 1's comments, I disagree with the change. I believe the original sentence provided by the authors better emphasized the importance of creating a local database. Including the number of sequences in the database detracts from the broader significance of this statement.

Line 134: consists of -> is located in

Line 135 - 140: Required English edition for clarity

Line: 148: suffering -> animal suffering

Line 150: and -> with

Line 159 -> minimize

Line 254: DNA databank -> database

Line 254-355: and that this -> which

Line 298: by -> relative to

Line 374: after -> following

Line 374: are identifiers used to -> serve as identifiers to

Line 392 - 393: the statement is correct however you should mention that the number of winter samples was limited

Reviewers' comments:

Reviewer's Responses to Questions

**Comments to the Author**

1. If the authors have adequately addressed your comments raised in a previous round of review and you feel that this manuscript is now acceptable for publication, you may indicate that here to bypass the “Comments to the Author” section, enter your conflict of interest statement in the “Confidential to Editor” section, and submit your "Accept" recommendation.

Reviewer #1: (No Response)

Reviewer #3: All comments have been addressed

2. Is the manuscript technically sound, and do the data support the conclusions?

Reviewer #1: Yes

Reviewer #3: Yes

3. Has the statistical analysis been performed appropriately and rigorously?

Reviewer #1: Yes

Reviewer #3: Yes

4. Have the authors made all data underlying the findings in their manuscript fully available?

Reviewer #1: Yes

Reviewer #3: Yes

5. Is the manuscript presented in an intelligible fashion and written in standard English?

Reviewer #1: Yes

Reviewer #3: Yes

6. Review Comments to the Author

Reviewer #1: I am almost satisfied with the author’s corrections to the revised manuscript. The manuscript has been much improved with appropriate explanations. The authors can still improve the points written below.

Minor concerns

Fig. 1: Exclude “include” in parentheses.

L446: analyzed region > analyzed DNA region

L446-447: There is a paper which showed the opposite trend: Mallot et el. (2018) https://doi.org/10.1371/journal.pone.0199556. This paper showed superiority of trnL over rbcL. It is not fair to show only one side view.

L456-458: This sentence is vague and therefore should be excluded because we cannot know the minimum quantity. The maximum volume of feces is limited by the DNA extraction kit (or method). It has been indicated that biological replication increased the obtained dietary taxa (Mata et al. 2019). That means that there we cannot estimate the total dietary taxa with feces less than 200 mg (usual maximum weight in the DNA extraction kit) and we need to apply biological replication with cost to know overall dietary trend.

Mata, V. A., Rebelo, H., Amorim, F., McCracken, G. F., Jarman, S. and Beja, P. 2019 How much is enough? Effects of technical and biological replication on metabarcoding dietary analysis. Molecular Ecology 28: 165–175.

L523-533: You can refer to Sato (2024) for the issue of contamination. If the review below is useful for your explanation, cite it because you did not cite any literature in this paragraph (therefore, it is not clear how the contamination issue is problematic).

Sato, J. J. 2024. Diets of rodents revealed through DNA metabarcoding. Mammal Study 50 (1): https://doi.org/10.3106/ms2024-0018.

Reviewer #3: (No Response)

7. PLOS authors have the option to publish the peer review history of their article (what does this mean? ). If published, this will include your full peer review and any attached files.

**Do you want your identity to be public for this peer review?** For information about this choice, including consent withdrawal, please see our Privacy Policy .

Reviewer #1: No

Reviewer #3: **Yes: ** Saeed Mohamadzade Namin

---

## [Author Response · Author response to Decision Letter 3]

24 Jan 2025

Dear Editor,

Thank you for your valuable comments. We have revised the manuscript in accordance with the reviewers' and editorial feedback

Dear Editor

Thank you for your useful comments. Responses to your comments are given below.

Editor comment E1:

Furthermore, while the authors revised lines 37–38 based on Reviewer 1's comments, I disagree with the change. I believe the original sentence provided by the authors better emphasized the importance of creating a local database. Including the number of sequences in the database detracts from the broader significance of this statement.

Response:

Revised manuscript: L36-39

We have corrected this sentence according to the Editors’ comment.

Editor comment E2:

Line 134: consists of -> is located in

Response:

Revised manuscript: L134

We have corrected this sentence according to the Editors’ comment.

Editor comment E3:

Line 135 - 140: Required English edition for clarity

Response:

Revised manuscript: L137-139, L144

This survey site is owned by some corporations, and there are no publications available from which vegetation information can be referenced. The vegetation survey results we have obtained represent the most detailed data available (see S1 Table).

The correction of this comment corrects Support Table 3 to 1, Support Table 1 to 2, and Support Table 2 to 3.

Editor comment E4:

Line: 148: suffering -> animal suffering

Response:

Revised manuscript: L149

We have corrected this sentence according to the Editors’ comment.

Editor comment E5:

Line 150: and -> with

Response:

Revised manuscript: L152

We have corrected this sentence according to the Editors’ comment.

Editor comment E6:

Line 159 -> minimize

Response:

Revised manuscript: L160

We have corrected this sentence according to the Editors’ comment.

Editor comment E7:

Line 254: DNA databank -> database

Response:

Revised manuscript: L256

We have corrected this sentence according to the Editors’ comment.

Editor comment E8:

Line 254-355: and that this -> which

Response:

Revised manuscript: L256

We have corrected this sentence according to the Editors’ comment.

Editor comment E9:

Line 298: by -> relative to

Response:

Revised manuscript: L299

We have corrected this sentence according to the Editors’ comment.

Editor comment E10:

Line 374: after -> following

Response:

Revised manuscript: L376

We have corrected this sentence according to the Editors’ comment.

Editor comment E11:

Line 374: are identifiers used to -> serve as identifiers to

Response:

Revised manuscript: L376

We have corrected this sentence according to the Editors’ comment.

Editor comment E12:

Line 392 - 393: the statement is correct however you should mention that the number of winter samples was limited

Response:

Revised manuscript: L393-394, L475

We have corrected this sentence according to the Editors’ comment.

Dear Reviewer #1

Thank you for your useful comments. Responses to your comments are given below.

Reviewer comment R1-1:

Fig. 1: Exclude “include” in parentheses.

Response:

Fig 1:

We have corrected this sentence according to the Reviewers’ comment.

Reviewer comment R1-2

L446: analyzed region > analyzed DNA region

Response:

Revised manuscript: L449

We have corrected this sentence according to the Reviewers’ comment.

Reviewer comment R1-3

L446-447: There is a paper which showed the opposite trend: Mallot et el. (2018) https://doi.org/10.1371/journal.pone.0199556. This paper showed superiority of trnL over rbcL. It is not fair to show only one side view.

Response:

Revised manuscript: L449-450

We have corrected this sentence according to the Reviewers’ comment.

Reviewer comment R1-4

L456-458: This sentence is vague and therefore should be excluded because we cannot know the minimum quantity. The maximum volume of feces is limited by the DNA extraction kit (or method). It has been indicated that biological replication increased the obtained dietary taxa (Mata et al. 2019). That means that there we cannot estimate the total dietary taxa with feces less than 200 mg (usual maximum weight in the DNA extraction kit) and we need to apply biological replication with cost to know overall dietary trend.

Mata, V. A., Rebelo, H., Amorim, F., McCracken, G. F., Jarman, S. and Beja, P. 2019 How much is enough? Effects of technical and biological replication on metabarcoding dietary analysis. Molecular Ecology 28: 165–175.

Response:

Revised manuscript: L461

We understand that the required weight of feces varies not only depending on the DNA extraction kit but also due to uncontrollable factors such as microbial content and the feeding amount. However, in the case of the analysis of Apodemus speciosus, it is true that there is variability in the fecal weight currently used for analysis. No studies on dietary resource analysis of this species have focused on fecal weight, and it is considered an important finding suggested by this research. Therefore, we would like to retain the expression as it is here.

This study�feces (ca. 60 mg)

Shirako et al. 2014: feces (50~500 mg)

Shirako et al. 2015: gastric contents (ca. 25–75 mg)

Sato et al. 2018, 2019: feces (only three fecal pellets)

However, since it is impossible to know the "minimum quantity" of feces required for the analysis, we modified it to "optimal amount".

Reviewer comment R1-5

L523-533: You can refer to Sato (2024) for the issue of contamination. If the review below is useful for your explanation, cite it because you did not cite any literature in this paragraph (therefore, it is not clear how the contamination issue is problematic).

Response:

Revised manuscript: L538-544

We have corrected this sentence according to the Reviewers’ comment.

And we added some references.

---

## [Decision Letter · Decision Letter 3]

29 Jan 2025

Identification of Large Japanese field mouse *Apodemus speciosus* food plant resources in an industrial green space using DNA metabarcoding

PONE-D-24-12099R3

Dear Dr. Fujii,

We’re pleased to inform you that your manuscript has been judged scientifically suitable for publication and will be formally accepted for publication once it meets all outstanding technical requirements.

Kind regards,

Faham Khamesipour, Ph.D.

Academic Editor

PLOS ONE

Additional Editor Comments (optional):

The manuscript can now be published.

Reviewers' comments:

Reviewer's Responses to Questions

**Comments to the Author**

1. If the authors have adequately addressed your comments raised in a previous round of review and you feel that this manuscript is now acceptable for publication, you may indicate that here to bypass the “Comments to the Author” section, enter your conflict of interest statement in the “Confidential to Editor” section, and submit your "Accept" recommendation.

Reviewer #1: All comments have been addressed

Reviewer #3: All comments have been addressed

2. Is the manuscript technically sound, and do the data support the conclusions?

Reviewer #1: Yes

Reviewer #3: Yes

3. Has the statistical analysis been performed appropriately and rigorously?

Reviewer #1: Yes

Reviewer #3: Yes

4. Have the authors made all data underlying the findings in their manuscript fully available?

Reviewer #1: Yes

Reviewer #3: Yes

5. Is the manuscript presented in an intelligible fashion and written in standard English?

Reviewer #1: Yes

Reviewer #3: Yes

6. Review Comments to the Author

Reviewer #1: I agree with your revision. Lastly, make a correction in the sentence below. "winter" appears three times in the sentence. Is it correct? I do not need your response. Just corect it.

L475-476: “in winter” should be deleted.

Reviewer #3: The authors of the manuscript have carefully addressed all the comments provided by the reviewers and made improvements to the paper accordingly. Each suggestion has been thoughtfully considered, and the necessary revisions have been implemented to enhance the clarity, quality, and scientific rigor of the manuscript.

7. PLOS authors have the option to publish the peer review history of their article (what does this mean? ). If published, this will include your full peer review and any attached files.

**Do you want your identity to be public for this peer review?** For information about this choice, including consent withdrawal, please see our Privacy Policy .

Reviewer #1: No

Reviewer #3: **Yes: ** Saeed Mohamadzade Namin

---

## [Editor Report · Acceptance letter]

PONE-D-24-12099R3

PLOS ONE

Dear Dr. Fujii,

I'm pleased to inform you that your manuscript has been deemed suitable for publication in PLOS ONE. Congratulations! Your manuscript is now being handed over to our production team.

Kind regards,

on behalf of

Dr. Faham Khamesipour

Academic Editor

PLOS ONE